# Continuous streamflow prediction in ungauged basins: Long Short-Term Memory Neural Networks clearly outperform traditional hydrological models

5    Richard Arsenault[1], Jean-Luc Martel[1], Frédéric Brunet[1], François Brissette[1], Juliane Mai[2]

[1]Hydrology, Climate and Climate Change Laboratory, École de technologie supérieure, 1100 Notre-Dame West, Montréal, Québec, H3C 1K3, Canada
[2]Department of Civil and Environmental Engineering, University of Waterloo, 200 University Ave W, Waterloo, Ontario, N2L 3G1, Canada

10    *Correspondence to*: Richard Arsenault (richard.arsenault@etsmtl.ca)

**Abstract.** This study investigates the ability of Long Short-Term Memory (LSTM) neural networks to perform streamflow prediction at ungauged basins. A set of state-of-the-art, hydrological model-dependent regionalization methods are applied to 148 catchments in Northeast North America and compared to a LSTM model that uses the exact same available data as the hydrological models. While conceptual model-based methods attempt to derive parameterizations at ungauged sites from other similar or nearby catchments, the LSTM model uses all available data in the region to maximize the information content and increase its robustness. Furthermore, by design, the LSTM does not require explicit definition of hydrological processes and derives its own structure from the provided data. The LSTM networks were able to clearly outperform the hydrological models in a leave-one-out cross-validation regionalization setting on most catchments in the study area, with the LSTM model outperforming the hydrological models in 93 to 97% of catchments depending on the hydrological model. Furthermore, for up to 78% of the catchments, the LSTM model was able to predict streamflow more accurately on pseudo-ungauged catchments than hydrological models calibrated on the target data, showing that the LSTM model's structure was better suited to convert the meteorological data and geophysical descriptors into streamflow than the hydrological models even calibrated to those sites in these cases. Furthermore, the LSTM model robustness was tested by varying its hyperparameters, and still outperformed hydrological models in regionalization in almost all cases. Overall, LSTM networks have the potential to change the regionalization research landscape by providing clear improvement pathways over traditional methods in the field of streamflow prediction in ungauged catchments.

## 1 Introduction

The ability to simulate streamflow at ungauged sites is a major unresolved problem in hydrology (Blöschl et al., 2019). Estimating flows in ungauged rivers, also known as streamflow regionalization, is a necessary step for many infrastructure

30    control projects such as flood control reservoirs, hydropower generation and management, and water availability for recreational, agricultural and environmental uses. Since the International Association of the Hydrological Sciences (IAHS)

2003-2013 decade on streamflow prediction in ungauged basins (PUB) (Sivapalan et al., 2003), numerous methods ranging from statistical (Castiglioni et al., 2011; Skøien and Blöschl, 2007) to conceptual or physical (Wagener et al., 2004; Wagener and Wheater, 2006) methods to transfer knowledge from gauged basins to those ungauged locations have been proposed. Some studies investigated the ability to estimate continuous streamflow time-series (Wagener et al., 2004; Zhang and Chiew, 2009), whereas others attempted to regionalize hydrological indices such as mean and peak flow values directly (Yadav et al., 2007; Zhang et al., 2018), foregoing the necessity to model the entire hydrograph.

In the past few years, a multitude of studies have documented the progress of regionalization methods, which attempt to solve, or alleviate, the problem of PUB. He et al. (2011); Hrachowitz et al. (2013); and Razavi and Coulibaly (2013) proposed detailed literature reviews following the IAHS decade on PUB, which readers are encouraged to consult for more in-depth knowledge about regionalization methods. In a three-part metastudy, Parajka et al. (2013); Salinas et al. (2013); and Viglione et al. (2013) analyzed the results of 34 regionalization studies. The aim of their study was to recommend best practices for regionalization based on the climatological and physiographic properties of the region of interest as well as the type of hydrological model. Here, and in this study in general, the term "hydrological model" refers to conceptual or physical models that represent hydrological processes and can simulate streamflow from meteorologic inputs. While some general trends were found, it was still generally recognized that more research needed to be performed to improve regionalization method performance. Guo et al. (2021) also evaluated the research effort in terms of regionalization across the globe since the end of the decade on PUB, and their compilation shows that this is still a very active field of research with novel methods being introduced continuously.

Artificial neural networks (ANNs) have long-been used in hydrology (e.g. Abrahart et al., 2012; Anctil and Rat, 2005; Coulibaly et al., 2000; Dawson and Wilby, 2001). Most studies used various versions of multilayer perceptron (MLP) networks and some applications of simple recurrent neural network (RNN) architecture. Despite a couple of decades of work, MLP networks have not been shown to outperform traditional conceptual/physical approaches in many different sub-fields, such as water quality, groundwater and streamflow modeling (Oyebode and Stretch, 2019), and method improvements are required for them to gain broader acceptance.

A Long Short-Term Memory (LSTM) network is a special type RNN introduced by Hochreiter and Schmidhuber (1997) that has built-in feedback connections that gives it the ability to learn sequence dependence. This property makes it particularly well suited to hydrological streamflow simulation/forecast problems where data series are typically strongly autocorrelated. LSTM models have the potential to reshuffle the modeling/forecasting landscape in the near future (Nearing et al., 2021). Even though LSTM models date back to 1997, their use has only recently drastically increased in the field of deep learning and particularly in the traditional artificial intelligence fields such as computer vision and speech processing (e.g. Guo et al., 2016; LeCun et al., 2015).

In hydrology, a breakthrough occurred with the seminal work of Kratzert et al. (2018, 2019a), which showed that deep learning (DL) could be successfully applied to streamflow modeling and regionalization studies, bringing significant improvements over previous state-of-the-art methods. In their study, Kratzert et al. (2019a) compared a modified Long Short-Term Memory (LSTM) model to accept inputs not only to be the meteorological forcing data, but also (static) catchment characteristics serving as inputs during training. Their model, named Entity-Aware LSTM (EA-LSTM), was then trained on a large number of catchments, using meteorological and physiographic properties to best represent streamflow at multiple gauges simultaneously. The EA-LSTM was then applied to ungauged catchments using only meteorological and physiographic properties of the ungauged basins. Their results showed that the EA-LSTM was generally able to provide better flow estimates at the ungauged sites than well-known conceptual hydrological models that were calibrated on observed streamflow data. This work showed the potential that LSTM models have in terms of representing the hydrological processes without having to make assumptions and hypotheses on their nature and structure.

Applications of deep learning in hydrology are all recent but most of the published studies have shown that neural networks outperform the baseline traditional approaches (Kratzert et al., 2018). The ability of LSTM models is now well established for streamflow modeling (Gauch et al., 2021a; Kratzert et al., 2019b) but they are also showing promises in other areas such groundwater modeling (Ali et al., 2022;Nourani et al., 2022), snow water equivalent mapping (Duan and Ullrich, 2021) and soil moisture modeling (Li et al., 2022). For example, Ayzel et al. (2021) used an LSTM model to generate a gridded runoff database in Western Russia that compared favorably to a similar dataset created using the GR4J hydrology model processes.

For streamflow modeling, in a traditional comparison mode, for which both the LSTM and hydrological models are trained/calibrated using only data from each single catchment, LSTM models have shown to globally outperform traditional conceptual and process-based hydrological models. In a study of 241 catchments from the CAMELS database (Addor et al., 2017), Kratzert et al. (2018) reported a mean NSE value of 0.63 for the LSTM model and 0.58 for the SAC-SMA conceptual lumped hydrological model in temporal validation. The latter only outperformed the LSTM model in the dry catchments of the Southwest of the United States, as the LSTM model suffered from the large portions of the streamflow time series being zero and hence only containing limited information. A similar performance gain was noted by Kratzert et al. (2019a) on a study with 531 catchments extracted from the same CAMELS database. Mean NSE values in out-of-sample regionalization were 0.69 for the LSTM model, 0.64 for the SAC-SMA conceptual hydrological model and 0.58 for the National Water Model, which is a process-based streamflow-generating model. The mean gain of 0.05 in terms of NSE efficiency between the LSTM and SAC-SMA conceptual model across both studies is quite significant in the field of hydrological modeling. However, the above results are only compared against two hydrological models, and the observed gain in efficiency might have been different (and possibly even be negative) if a different hydrological model structure had been chosen as the baseline comparison model. Feng et al. (2020) showed that using regional flow duration curves as predictors in an LSTM model improved the prediction in ungauged basins skill over 671 CAMELS basins compared to an LSTM model without the flow duration curve inputs. The

Mai et al. (2022) study specifically looked at this issue by comparing 12 hydrological models locally and regionally calibrated against one globally trained LSTM model over the Great Lakes watershed. The LSTM model was the best performing one across all validation experiments (temporal, spatial, spatio-temporal) regarding streamflow (with an improvement in median KGE compared to the second-best model of 0.03, 0.15, and 0.15, respectively).

The LSTM models particularly shine when they are globally calibrated using data from all catchments. In the Kratzert et al. (2018) study, the regional LSTM models (single models that can predict streamflow on a variety of catchments in a region) performed on average just as well as the local LSTM (trained specifically on a single catchment at a time) with a median NSE difference of 0. Ayzel et al. (2020) also demonstrated that for 200 catchments in Northwest Russia, LSTM-based regionalization outperformed the regionalized GR4J model with median NSE values of 0.73 and 0.61, respectively.

Furthermore, Nearing et al. (2021) showed that the median improvement of regionally trained LSTMs improved the performance of single-basin trained LSTM by a median value of 0.10 in NSE performance by using basins used in Kratzert et al. (2019b). This is in sharp contrast to regionalization using a hydrological model in which a performance loss is unavoidable since a hydrological model calibrated on a dataset has access to all of the information, and this information asymmetry guarantees a better performance for the calibrated models over the regionalized ones. This shows that LSTM models are very

efficient at extracting information from large datasets (Kratzert et al., 2022; Ayzel et al. 2020; Kratzert et al., 2019a; Kratzert et al., 2019b), displaying the added value of deep learning approaches. It also shows that large hydrological datasets contain more information that can possibly be extracted by conceptual and process-based hydrological models (Nearing et al., 2021). Choi et al. (2022) showed that this was also the case in Korea, where their LSTM regionalization implementation was performed over 13 catchments with satisfactory results but that could likely be improved with access to more catchment data.

These results also seem to hold in more arid climates, according to Nogueira Filho et al. (2022), who showed that both LSTM and feed-forward neural networks outperformed conceptual hydrological models for regionalization in a semi-arid region of Brazil. Zhang et al. (2022) were able to obtain better results than with conceptual hydrological models using a regional encoder-decoder LSTM on 35 catchments in China for streamflow prediction in ungauged catchments, showing that this approach can be used with success even with limited numbers of training catchments.

Downsides of LSTM models (and of all deep learning methods) is the need to rely on large datasets for training the artificial neural networks and potential difficulties at extrapolating extremes in conditions outside the range of existing datasets. Kratzert et al. (2018) showed that for daily streamflow simulation, 15 years is a lower bound for the training, with additional years also needed as a validation period (not to be confused with the testing period also used in machine learning). Gauch et al. (2021a);

and Gauch et al. (2021b) also studied the impact of the length of training data and showed a performance loss for shorter periods. Ayzel and Heistermann (2021) came to the same conclusion and compared the incremental performance gain obtained by adding additional training years to the GR4H conceptual hydrological model and two types of RNNs, i.e., LSTM and GRU (Gated Recurrent Units). Results showed that the temporal validation performance of GR4J rose rapidly and transitioned to an

asymptotic behavior on an independent 9-year validation period after just three years of streamflow data used for calibration, whereas the performance in validation of both deep learning approaches rose much slower and did not reach an asymptotic plateau even when using the full 14-year calibration period. On half their test catchments, the RNNs performed better than GR4H after training on a 14-year window, and for the other catchments the RNNs still performed acceptably well. Other data requirements for regionalization include catchment physical descriptors. Although some are relatively easy to evaluate (land cover, slope, elevation, area, etc.), others can be more difficult (e.g., soil depth, soil porosity, etc.). Li et al. (2022) recently showed that it was possible to train the LSTM model in regionalization using random vectors of descriptors for those unavailable, while still maintaining similar levels of performance as if the original descriptors were all available. With respect to the extrapolation to extreme events problem, Frame et al. (2022) showed that LSTM models are relatively accurate at producing high flows when compared to SAC-SMA and a process-based model (US National Water Model), even when extreme events were excluded from the training. This suggests that LSTM models are able to not only extract relevant hydrological information from the training dataset but actually learn from it. This idea was explored by Lees et al. (2022) who showed that LSTM models do indeed have the capability of learning, and that these learned representations can even be interpreted by scientists into process understanding.

Ultimately, the above studies are consistent in the finding that LSTM models perform as well (worst case scenario) or better than traditional approaches. The ability of LSTM models at using large datasets from multiple catchments makes them particularly well suited for prediction at ungauged basins; a fact that has been underlined by a few studies (e.g., Kratzert et al., 2019a; Kratzert et al., 2019b; Ayzel et al., 2020; Mai et al., 2022). However, actual performance in a regionalization context has only been indirectly assessed based on the performance of regional LSTM models compared against that of hydrology models specifically calibrated at each catchment, or in some cases against a regionally calibrated hydrology model. The former is a fair assessment since a performance loss is inevitable in a regionalized setting. However, the fact remains that LSTM models have yet to be compared against traditional regionalization approaches in a true "leave-one-out cross-validation" (LOOCV) setting using state-of-the-art hydrological model-based methods on ungauged basins.

Thus, the first objective of this work consists in comparing the performance of a LSTM model against that of traditional state-of-the-art regionalization methods based on hydrological models. The second objective consists in an analysis of the strengths and weaknesses of the developed LSTM model and to propose avenues of research to improve its performance.

## 2 Data and study area

The study area is located in Northeast North America and is composed of 148 catchments that are at least occasionally subjected to snow accumulation and melt events. The catchments were taken from the HYSETS database (Arsenault et al. 2020) that contains over 14,000 catchments including flow and meteorological data over North America. The subdomain

shown in Fig. 1 was selected to maintain reasonable computing time and memory management. The spatial extent, as shown in Fig. 1, ensures that the catchments in the southern part of the domain have a hydrological signature that strongly differs (e.g., earlier peak flow, a much larger fraction of liquid precipitation) than those in the north, which therefore requires modeling of the processes and cannot simply rely on simple transfer functions based on catchment area (e.g., Fry et al., 2014). Furthermore, only catchments with a drainage area of more than 500 km$^2$ were included in this study, to avoid scale and lag issues when regionalizing with hydrological models calibrated on a daily time step.

[FIGURE 1 HERE]

Table 1 presents the main properties of these catchments which will be used as descriptors for the regionalization methods described below. These were taken directly from the HYSETS database (Arsenault et al., 2020), therefore only a summary is presented here. In HYSETS, land cover data was computed from the North American Land Cover Monitoring System (NALCMS) of 2010, whereas slope, aspect and elevation were computed from the EarthENV 90-meter Digital Elevation Model (Robinson et al., 2014). Climatological indicators were computed from the meteorological data directly (as discussed below), and the aridity index was computed from the observed precipitation and estimated potential evapotranspiration as computed by the Oudin et al. (2005) method. All catchment descriptors were averaged at the basin scale. It should be noted that there exist a multitude of possible descriptor sets to use, such as that in Table 4 in Kratzert et al. (2019b) which has shown good results. Some descriptors are included as they are required for hydrological model-based regionalization, such as longitude and latitude, that are typically used as proxies for unknown data that are assumed to be spatially relatively homogeneous such as soil properties and bedrock depth.

**Table 1: List of catchment descriptors used in the regionalization experiment. All values are spatial averages over each basin. The descriptors were derived based on the EarthENV DEM, the NALCMS land cover database, and ERA5 reanalysis meteorological data.**

| Catchment descriptors (unit) | Minimum | Median | Maximum |
|---|---|---|---|
| Area (km$^2$) | 500.8 | 1149.6 | 31900.0 |
| Longitude (°E) | -84.9 | -75.8 | -62.0 |
| Latitude (°N) | 41.1 | 44.2 | 49.9 |
| Elevation (m) | 75.9 | 348.2 | 722.5 |
| Slope (%) | 0.3 | 3.5 | 12.2 |
| Aspect (°) | 3.8 | 163.5 | 355.7 |
| Gravelius (-) | 1.3 | 1.9 | 3.7 |
| Perimeter (km) | 113.5 | 233.5 | 1846.2 |
| Land cover - Crops (%) | 0.0 | 8.9 | 86.0 |
| Land cover - Forest (%) | 4.4 | 73.2 | 96.6 |
| Land cover - Shrub (%) | 0.0 | 1.7 | 14.7 |

| | | | |
|---|---|---|---|
| Land cover - Grass (%) | 0.1 | 0.9 | 8.6 |
| Land cover - Water (%) | 0.0 | 1.0 | 13.9 |
| Land cover - Wetlands (%) | 0.0 | 2.7 | 19.6 |
| Land cover - Urban (%) | 0.1 | 3.6 | 64.2 |
| Permeability ($m^2$) | -16.5 | -14.5 | -11.8 |
| Porosity (%) | 1.0 | 12.4 | 23.7 |
| Mean annual precipitation (mm) | 814.7 | 1153.7 | 1432.1 |
| Mean annual evapotranspiration (mm) | 435.4 | 637.3 | 798.3 |
| Mean snow water equivalent (mm) | 2.0 | 16.8 | 109.9 |
| Aridity index (-) | 0.35 | 0.54 | 0.80 |
| High precipitation frequency (ratio of number of days with precipitation > 5x average precipitation over total number of days) (-) | 0.03 | 0.05 | 0.06 |
| Low precipitation frequency (ratio of number of days with precipitation < 1mm over total number of days) (-) | 0.47 | 0.58 | 0.66 |
| High precipitation duration (average number of consecutive days with precipitation > 5x average precipitation) (days) | 1.07 | 1.10 | 1.16 |
| Low precipitation duration (average number of consecutive days with precipitation < 1mm) (days) | 2.35 | 2.89 | 3.43 |

The meteorological data were taken from the HYSETS database, which contains data for over 14,000 catchments in North America (Arsenault et al., 2020). Various data sources are available in HYSETS, but for this study, the daily ERA5 reanalysis data (Hersbach et al., 2020) was preferred since there are no missing values and multiple studies have shown its reliability for hydrological modeling over the study domain (Tarek et al., 2020a, b). Meteorological data (rainfall, snowfall, minimum and

195 maximum temperature) cover the period 1979 to 2018 inclusively, on a daily time step. Daily streamflow data were also taken from the HYSETS database, which aggregated daily flow data from Environment and Climate Change Canada (ECCC)'s Water survey Canada (WSC) and the United States Geological Survey (USGS). Flow data covers the period ranging from 1979 to 2018 inclusively, with many stations being only available on subsets of that period, typically having between 10 and 20 years of available data. However, in this study, only catchments that had at least 30 years of data (even if not sequential)

were preserved, guaranteeing that each catchment has a long enough observational record for results to be robust and representative. After this filter was applied, 148 catchments remained over the study domain in Northeast North America.

**3 Methods**

The methods can be separated into three main themes: Hydrological model preparation and calibration (Sect. 3.1), "classical" regionalization method application, using hydrological models (Sect. 3.2), and creation and application of the LSTM model

applied to the problem of prediction of streamflow in ungauged basins (Sect. 3.3).

## 3.1 Hydrological models and calibration

Three lumped hydrological models were implemented for the model-based regionalization. These are models that were previously used in streamflow regionalization studies in the similar region, i.e., in the province of Québec, Canada. The first is the HSAMI model, which was used in Arsenault and Brissette (2014) over 268 catchments in Quebec, Canada. HSAMI is used by Hydro-Québec in operational forecasting for hydropower management (Fortin, 2000). It is a conceptual model which has 23 calibration parameters (see Table S1). It simulates infiltration, runoff, evapotranspiration, snow accumulation and melt, and flow routing, and contains three storage reservoirs, representing surface flow, vadose zone flow and saturated zone flow. Water is routed to the outlet using two unit hydrographs. The second is the HMETS model (Martel et al., 2017), which was implemented for regionalization in the same study area as the previous study in Arsenault and Brissette (2016). HMETS is a conceptual model that contains 21 parameters (see Table S2) and has a more complex snow model than HSAMI, but also has a less complex infiltration and routing setup. The final model is the GR4J model (Perrin et al., 2003), which is widely used across the world in hydrology studies and was also implemented in regionalization over the province of Quebec in Poissant et al. (2017). GR4J is a simple, 4 parameter model that simulates the rainfall-runoff process using 2 storage reservoirs and a unit hydrograph-based routing scheme. However, it does not simulate snow processes, therefore the CemaNeige snow model (Valéry et al., 2014) was added to account for snow accumulation and melt. This also added two parameters, for a total of 6 parameters for the GR4J-CemaNeige (GR4JCN) model (see Table S3). The rationale of using GR4JCN is that there are few parameters, thus it is more likely that parameters will be linked to a physical process due to lesser equifinality. This should improve the relationship between parameter values and physical response of the model, which can be seen as an advantage for PUB studies.

Each of these models was calibrated using the Covariance Matrix Adaptation Evolution Strategy (CMAES; Hansen et al., 2003) optimization algorithm. They were used and calibrated with good results in the Arsenault and Brissette (2014) (HSAMI), Arsenault and Brissette (2015) (HSAMI, HMETS) and Poissant et al. (2017) (GR4JCN) studies, and model parameter bounds are reused here to maintain the same level of model flexibility. Calibrations were performed using an upper limit of either 5000 (GR4JCN) or 10,000 (HSAMI, HMETS) model evaluations. The objective of the calibration was to maximize the Nash Sutcliffe Efficiency metric (NSE; Nash and Sutcliffe, 1970). NSE was selected despite the Kling-Gupta Efficiency (KGE; Gupta et al., 2009) metric being better suited, solely to ease comparisons between the previous regionalization studies that were strongly reliant on NSE and this current study. All hydrological models were calibrated on the entire period of 1979-2018 as suggested by Arsenault et al. (2018); and Shen et al. (2022) , while keeping the first available year (1979) of each catchment as the warmup period.

[FIGURE 2 HERE]

Calibration results for the three models are shown in Fig. 2, in which results for calibration over each of the 148 catchments

are shown in boxplots. It can be seen that most catchments display acceptable to strong NSE values, with a median NSEs of 0.67, 0.67 and 0.68 for the GR4JCN, HMETS and HSAMI models, respectively. Some catchments display calibration results below 0.5 for some models. These were kept in the study to evaluate how they can impact the regionalization results, as described in the following section.

**3.2 Model-dependent regionalization methods**

The hydrological models were used as the transfer functions to estimate flows on the ungauged sites based on the meteorological and physiographic properties. A suite of six regionalization methods was implemented for each model. Regionalization skill was evaluated using a leave-one-out cross-validation (LOOCV) approach, by which each catchment was in turn considered as (pseudo-)ungauged while the regionalization approaches were applied to try and estimate its flows

(Parajka et al., 2005). This allowed performing 148 regionalization tests for each of the 18 scenarios, i.e., the combination of hydrological model (here 3) and regionalization method (here 6).

In this study, two well-known and omnipresent regionalization methods were implemented: the spatial proximity method and the physical similarity method. For both cases, some variants were introduced to increase performance as recommended by

255 various studies in the literature (He et al., 2011; Oudin et al., 2008; Razavi and Coulibaly, 2013). The well-known methods use the following framework:

1) All available catchments are modeled and calibrated using a hydrological model in order to prepare optimal parameter sets independently for all basins, but only the results of the N-1 (all except the one considered ungauged) basins are
260 used for regionalization at the pseudo-ungauged site.
2) The most similar (physical similarity method; PS) or closest (spatial proximity method; SP) catchment is considered as the "donor" catchment. The donor catchments' calibrated parameters are transferred to the ungauged site. Here, the most (physically) similar catchment refers to the catchment that has the smallest absolute difference between all the standardized catchment descriptors to ensure equal weighting of each descriptor. The (spatially) closest catchment
is the one whose centroid is nearest in the latitude/longitude domain.
3) The ungauged basin is set up using meteorological forcings, static catchment attributes, and the donor catchment parameters. This setup is then used to simulate streamflow using the observed meteorological data.
4) The simulated streamflow at the pseudo-ungauged site is then compared with the observations, and the NSE score is computed.

This process is repeated for each of the 148 catchments and all three models. Furthermore, to improve performance, some simulations were performed with variations on the standard regionalization approaches:

A) Multi-donor simulations: In this case, more than one donor is used, such that the $N$ nearest catchments transfer their parameter sets to the ungauged catchment, and streamflow is generated for each case, resulting in $N$ simulated hydrographs. The average of these hydrographs is then taken, resulting in a single, more accurate hydrograph. This has been demonstrated in many studies, with between $N=4$ and $N=8$ donors being recommended as the optimal value (Arsenault and Brissette, 2014; Oudin et al., 2008). In this study, all tests performed used the multi-donor approach with $N=5$ donors as a generally accepted good compromise value without expending computing resources on larger donor sets.

B) Inverse-distance weighting (IDW): IDW is a variant of multi-donor simulations. In this case, the averaging of multi-donor hydrographs is performed according to the degree of similarity (or distance) between the donors and the ungauged site using weights

$$w_i = 1 - \left( \frac{d_i}{\sum_{j=1}^{N} d_j} \right),$$
(1)

Where $d_i$ is the distance/similarity between donor basin $i$ and the ungauged basin, and $N$ is the total number of donor basins in the weighting (here $N=5$ in this study). Therefore, more similar catchments are weighted more heavily in the hydrograph averaging. This has also been noted as a significant improvement over standard multi-donor regionalization (Arsenault and Brissette, 2014; Oudin et al., 2008; Parajka et al., 2005).

C) Removal of poor donor catchments: In regionalization, if a donor catchment is of poor quality (data quality problems, unreliable parameter set, etc.), then it can be considered unreasonable to use it as a donor catchment for other sites. In this study, tests were performed both with and without this filter to evaluate its impact. Even "poor" donor catchments were used as targets in the LOOCV. However, they did not contribute as donor catchments for other sites in this scenario. The filter applied here was the same as in Arsenault and Brissette (2014), which is the removal of all catchments whose calibration NSE was below 0.7. This translates to 84 to 89 basins being considered "poor" depending on the hydrology model (see Figure 2). Here, the multi-model method still always uses $N=5$ donors but these exclude the "poor" donors.

Overall, the tests performed for each hydrological model and both the spatial proximity and physical similarity methods, using multi-donor averaging ($N=5$ in all cases), are:

I.    Regular proximity/similarity approach with no poor catchment filter (ALL),

II.   Regular proximity/similarity approach with a selection using a poor catchment filter (SEL),

III.  Regular proximity/similarity approach with IDW and poor catchment filter (SEL IDW).

The results of these six methods were then compared with the LSTM model approach, detailed in the next section.

### 3.3 LSTM regionalization model

A Recurrent Neural Network (RNN) is a type of artificial neural network that can be used for the prediction of time series. As highlighted by Bengio et al. (1994), simple RNN have difficulty remembering information for long periods of time. For hydrological modeling application, this is problematic considering the need to track state variables up to multiple weeks or

310     months, such as the snow water equivalent within the snowpack or soil moisture. Long Short-Term Memory (LSTM) is a variant of an RNN that has been introduced by Hochreiter and Schmidhuber (1997) which allows the tracking of long-term dependencies between input and output sequences. Kratzert et al. (2018) and Kratzert et al. (2019a) offer a detailed description of the working behind an LSTM unit. Note that a LSTM model is composed of multiple LSTM units that can be interconnected in multiple layers.

In this study, the network architecture used is composed of two main branches for: 1) dynamic inputs fed into 2 LSTM layers each with 512 units followed by a dropout of 0.3, and 2) static inputs fed into a 25 neurons dense layer with a dropout of 0.1 followed by a leaky Rectified Linear Unit (ReLU) activation function. The leaky ReLU was used instead of a regular ReLU to eliminate any possibility of generating impossible objective function values or exploding gradients, which could sometimes

appear depending on the training convergence and learning rate. This provided more robustness and a bit more model flexibility at the expense of a small amount of extra computing time. Outputs from these two branches are then concatenated into a 20 neurons dense layer, activated with a regular ReLU function before being fed into a final one neuron dense layer. This setup is similar to that in Kratzert et al. (2019b) but adds an extra LSTM layer and doubles the number of LSTM units per layer. In their paper, Kratzert et al. (2019b) tested multiple model structures and hyperparameters, including up to 256 units per LSTM

layer, both for one and two layers, and with dropout rates ranging from 0.0 to 0.5 and input sequence lengths of 90 to 365 days. They finally settled for the model that provided the highest median, which was a single-layer, 256-unit LSTM with a dropout rate of 0.4 and an input sequence length of 270 days. However, the static descriptors were directly embedded in the LSTM layers, as opposed to their addition in a separate, parallel branch that is also tuned during training in this study. The model structure can be visualized in supplementary materials Fig. S1. The codes are made available at the location indicated in the

data availability statement below. Simpler models are also tested and discussed in section 5.2.

The LSTM model used in this study is designed to only predict one day of streamflow at a time, following the previous 365 days of the four following dynamic variables: rainfall, snowfall, minimum and maximum temperature. This is repeated $T$ times to create a simulation of streamflow of length $T$. The 25 static descriptors presented in Table 1 allow the model to distinguish

between each catchment.

To improve the learning, the following preprocessing of the data was conducted. Static descriptors were normalized between 0 and 1 using a min-max scaler, while the dynamic variables (meteorologic data) were standardized by the mean and the

standard deviation, which is a standard practice. Both scaling operations were performed on the training catchments (i.e. all except the ungauged catchment) only, leaving both validation catchments and testing catchments out of the process to fit the scalers' parameters to avoid contaminating the scaler with information it is not supposed to have access to. Once the scaler is trained, the scaling is applied to the validation and testing catchments. The target variable of streamflow was not itself scaled using this approach, since the model output and target values are not part of the model training computations and thus have no impact on the obtained results. Instead, the specific streamflow was used as the target variable by dividing streamflow records by the drainage area and converting to $mm.d^{-1}$. This was done to allow combining information from the multiple training catchments during the LSTM training since all streamflow values were now represented in an area-independent depth unit, while at the same time ensuring all values had similar magnitudes to avoid convergence problems. Without this mechanism, larger catchments and their larger flows would be weighed more heavily in the NSE objective function. In regionalization, the LSTM model would then output the same units (i.e., $mm.d^{-1}$), which could then be multiplied by the pseudo-ungauged basin's area and converted back to $m^3.s^{-1}$.

The training and application of the LSTM models to regionalization used the following setup. First, the pseudo-ungauged catchment (e.g., testing) is first identified out of the 148 catchments. In LOOCV, each catchment will eventually be considered the pseudo-ungauged catchment. Thus, from the 148 catchments, 147 remain. Of these, 80% (118 catchments) are randomly selected to be used as the training dataset. These will provide the catchment descriptors and meteorologic data to train the LSTM model. This leaves 20% (29 catchments) as the validation dataset. The validation dataset is not used for training except to act as a stopping criterion for the training operation. This is to prevent any overfitting during training. While these 29 catchments are not used directly for training, they are also not independent enough to evaluate the model performance. Therefore, testing occurs only on the pseudo-ungauged basin, and these testing regionalization metric values are those that are presented in this study.

### 3.4 Hyperparameter selection

Furthermore, to evaluate the sensitivity of the results to the LSTM model structure, multiple structures were tested, from simple single-layer LSTMs with 128 units, to complex dual-layer LSTMs with 512 units each, and in each case using various combinations of dense layers for the static inputs, activation functions, regularization options, batch sizes and objective functions. In all cases, a decaying learning rate was implemented to ensure proper convergence of the training algorithm, refining the learning rate step as a function of the number of epochs. The type and dimensionality of each element selected to build the network structure all play a role in the model performance and robustness, but through trial and error, a generally stable setup (one that led to good performance throughout the trials in LOOCV) was found and implemented. However, to analyze this point further, a series of tests was performed by repeating this study with an array of varying model hyperparameters. Nine additional runs were performed using the same general structure but with the adaptations as shown in Table 2.

**Table 2: LSTM hyperparameter variations used to evaluate the model structure robustness**

| Run ID | Training window length (days) | LSTM units | Other notes |
|---|---|---|---|
| 1 | 365 | 128 | Simplest LSTM model in this study |
| 2 | 365 | 2 x 128 | Simplest 2-layer LSTM model |
| 3 | 270 | 256 | Uses a shorter data window for training |
| 4a-4e | 365 | 256 | LSTM model repetitions with only the random seed changed for uncertainty analysis |
| 5 | 365 | 2 x 256 | 2 layers of LSTMs, each with 256 LSTM units |
| 6 | 365 | 512 | - |
| 7 | 365 | 2 x 512 | Base case considered and used as the comparison for the hydrological models. 2 layers of LSTMs, each with 512 LSTM units. Most complex LSTM model in this study |
| 8a-8b | 365 | 256 | Same model as 4a-4e but removing the catchments that were most difficult to train on to simulate the removal of "bad" catchments as performed with the hydrological model regionalization. Two repetitions with different initial seeds are performed |

Each of these modes was used as the LSTM model and results were compared to the hydrological model-based regionalization approaches. For the following results, model #7 (2-layer, 512-unit LSTM model) was implemented.

## 4 Results

The first step in assessing the LSTM performance was to compare its ability to simulate flow at the pseudo-ungauged sites to that of the hydrological models. This was done in two steps. First, the LSTM results in leave-one-out cross-validation (LOOCV) were compared to the hydrological models calibrated at the individual sites. This gives a significant advantage to the hydrological models, as they are directly calibrated on the available streamflow data and hence have access to all the information. The LSTM, on the other hand, only has access to the streamflow data from all but the pseudo-ungauged basin, therefore it does not have access to the target catchment streamflow in this step. These results are shown in Fig. 3 comparing the three hydrological models (x-axis) to the LSTM results (y-axis).

[FIGURE 3 HERE]

Results show that the LSTM is able to perform surprisingly well considering the information asymmetry compared to the hydrological models. The LSTM is able to perform at least as well as the GR4JCN, HMETS and HSAMI models in 75%, 78% and 73% of basins, respectively. The reasons for this will be discussed in section 5.1.

The second step was to compare the LSTM in LOOCV to all three hydrological models in regionalization (i.e., spatial validation), putting both model categories on the same playing field. Fig. 4 presents the overall results of the hydrological model performance when using various regionalization methods to obtain results at (pseudo-)ungauged locations. The

performance of the LSTM model is shown as well (same results as presented for LSTM in Fig. 3). Furthermore, it presents the
maximum skill attained by any of the 18 hydrological model and regionalization method combinations over each catchment,
as a best-case scenario for the hydrological model group (BEST HM). BEST-HM is a utopic case in that it would not be
possible ahead of time to determine which model or regionalization method would be the best, therefore it serves only to show
the best possible outcome the models and regionalization methods could provide.

[FIGURE 4 HERE]

From Fig. 4, it is clear that the choice of the hydrological model plays only a small role in the regionalization performance,
while the selection of the regionalization method plays a more significant role. Removing poorly calibrated catchments did
not increase overall regionalization skill, contrary to Oudin et al. (2008); and Arsenault and Brissette (2014). However, all
three conceptual hydrological models fall short of the performance of the LSTM, which displays a median NSE of 0.78,
compared to values ranging from 0.58 to 0.63 for all of the 18 tested configurations. The LSTM model's median NSE of 0.78
is also better than BEST-HM (median NSE of 0.66). While useful to help interpreting the results in the context of this study,
it is nonetheless important to stress that in a real-world application to an ungauged basin, the BEST-HM approach is not
feasible. However, compared to the LSTM, it can be seen that the distribution of NSE values is significantly inferior. When
each method is directly compared to the LSTM model (values added as labels on top x-axis in Fig. 4), it can be seen that the
LSTM model outperforms each of the 18 model-regionalization-approach combinations in at least 93% of the studied basins.
The LSTM outperforms the BEST-HM in 86% of the basins.

In an attempt to explain the higher performance of the LSTM, a few metrics were analyzed. First, the relationship between the
LSTM testing NSE (in LOOCV) and each catchment descriptor was evaluated, as presented in Fig. 5.

[FIGURE 5 HERE]

It can be seen from Fig. 5 that most catchment descriptors have little to no linear correlation to the LSTM testing NSE. There
is no notable structure, which seems to indicate that the LSTM does not favor one type of catchment over another. In essence,
this points to the LSTM being robust over the study area and makes it more likely to be applicable to other ungauged basins
in the study domain or having similar hydroclimatological and geomorphological properties, supporting results from Fig. 4.

The results in Figs. 3 and 4 show aggregated distributions but do not provide specific information for a comparison at each
site. To allow visualizing the spatial distributions of the methods' performance, maps (Fig. 6) and scatter plots (Fig. 7) of the
regionalization method leading to the best median for each hydrological model as well as the difference between those and the
LSTM in testing mode are shown hereafter.

[FIGURE 6 HERE]

[FIGURE 7 HERE]

Figure 6 again emphasizes the overall superior performance of the LSTM model (right column panels) while no clear spatial patterns can be detected in terms of hydrological model performance (left column panels). Figure 6 also shows that some

catchments perform poorly in regionalization for all hydrological models but have large improvements when using the LSTM model. The exact opposite is observed on a few catchments, and reasons for this are given in section 5.2.

Overall, the results indicate that the LSTM outperforms the best regionalization methods on 93%, 97% and 95% of the catchments for GR4JCN, HMETS and HSAMI, respectively (Figs. 4 and 7). This is important, considering that a strong

performing hydrological model with the best regionalization method is still outperformed on average by a LSTM that, while internally and structurally quite complex, required very little work to setup and train compared to the work required to setup, calibrate, and regionalize the hydrological models on the 148 donor catchments.

Finally, the sensitivity of the hyperparameter selection (i.e., LSTM model structure in this case) is shown in Fig. 8. It can be

seen that the results generally increase in performance with more complex model structure. They are also all better than the hydrological model-based regionalization methods (see Fig. 4). The only exception is that of the 8a-8b models with removal of catchments that were more difficult to train on for the general training, which strongly impacted results. Nonetheless, even these sub-optimal LSTM parameterizations outperform the hydrological model-based regionalization methods.

[FIGURE 8 HERE]

The five runs that were repeated to assess variability due to the model random state also showed small variability, which seems to show that the LSTM does not always converge to the same parameterization for its weights and biases. It is also important to note that the training (80%) and validation (20%) basins are categorized as such randomly, so the training step is performed

on different catchments for each of the 5 runs #4a-#4e. This can also explain a large part of the variance in the results within this sub-group. Also, it seems that filtering out the poor catchments, i.e., those that displayed the lowest training NSE values when training over the entire dataset, caused many basins to underperform and is not recommended.

**5 Discussion**

This section will discuss the regionalization results with the hydrological models and the LSTM model in section 5.1, as well
as the LSTM structure and hyperparameter selection in section 5.2.

**5.1 Comparison of hydrological model-based and LSTM regionalization**

This study confirms the recent trends in the literature according to which LSTMs are able to predict streamflow in ungauged
basins with performance levels competing or surpassing that of hydrological model-based methods. The results obtained in
this study show that over 148 catchments, the LSTM was able to clearly outperform six regionalization approaches based on
results derived from three different hydrological models on almost all catchments (Fig. 4). Furthermore, the LSTM was able
to simulate streamflow better than specifically calibrated hydrological models for many catchments (Figs. 3 and 6), further
demonstrating the potential skill in applying an LSTM model not only in regionalization studies, but in hydrological modeling
overall. Indeed, the well-trained LSTM was able to simulate streamflow better than hydrological models that had access to the
streamflow observations, which implies that the LSTM was able to build relationships that were more accurate than those
programmed in the hydrological models themselves. This also means that the very flexible LSTM framework can be adapted
to various regions and automatically train its weights on new data to represent different physical processes.

Therefore, besides its apparent versatility and performance, the LSTM also has the advantage that it removes one large problem
in classical model-based regionalization, which is the necessity to select the best donor catchment(s). The LSTM simply ingests
all the available information and then builds its internal structure from the data to match the observations. This is a significant
advantage, since it can be seen in Fig. 4 that the choice of a regionalization method plays an important role and the number of
donors to use also has an impact on regionalization model performance (Arsenault and Brissette, 2014). Other regionalization
methods often use multiple linear regression techniques to link parameter values to catchment descriptors (Oudin et al., 2008)
or use kriging to interpolate parameters over the spatial domain (Parajka et al., 2005). These methods are limited by the type
of relationship they can model. Furthermore, having a wide array of catchments could possibly lead to difficulties in modelling
parameter-descriptor relationships. For example, in this study, one catchment has a much larger area than almost all the others.
For a hydrological model-based regionalization approach, this might skew the regressions between catchment descriptors and
model parameters. LSTMs, on the other hand, are strongly non-linear and are thus not bound to these limitations. They could
also use these data to better predict streamflow processes at scales between the small and large catchments. This is because
neural networks in general, including LSTM-based neural networks, are particularly good for interpolating within the domain
they are trained to represent but can be unpredictable while extrapolating outside of the parameters of their training dataset.
Therefore, adding catchments with a wide array of properties confers the ability to establish relationships that other methods
simply cannot attain by widening the domain on which the model can interpolate. Furthermore, for the similar proximity and
physical similarity methods, the driving hypothesis is that more similar (or nearer) catchments should also behave similarly in

hydrological terms. However, Oudin et al. (2010) showed that this was not necessarily the case. The application of LSTMs to regionalization thus also bypasses this hypothesis completely, which is another significant advantage. Finally, on a related note, the parameter identifiability is made more difficult as more catchment predictors are included. Determining which should be used can have an impact on the hydrology model regionalization methods. However, the LSTM model can automatically parse and adjust the weights of catchment descriptors, discarding (or heavily reducing the weight of) descriptors with little

predictive power.

    However, LSTMs are also limited and disadvantaged in a few aspects. First, the nature of the LSTM model makes it extremely difficult or practically impossible to follow the data through the model and to attempt to identify relationships between model states and physical processes, or to determine why the model predicted a certain value of streamflow. With hydrological

models, each physical quantity and flux of the water cycle is estimated and can be tracked through time to determine if any problems occur. They also allow users to extract diagnostic variables during the simulation to evaluate other hydrological variables at the ungauged site, such as snow water equivalent, groundwater storage and other such variables of interest. Therefore, the LSTM can only estimate values it was trained on. Recent studies (Lees et al., 2022; Kratzert et al., 2019a; and Kratzert et al., 2019b) have shown that for an LSTM trained on a catchment, it was possible to derive hydrological processes

from the states and weights of the LSTM model. This has yet to be applied to a large-sample LSTMs in regionalization, but it is possible that some research will elucidate this in the near future. Furthermore, some studies have started investigating the possibility of adding physical constraints within the LSTM structure (such as ensuring mass-balance) (Frame et al., 2022; Hoedt et al., 2021), which might pave the way to a better understanding of the underlying relationships built within the LSTM structure.

Another limitation is the need for long observation data time series to adequately train the LSTM models compared to traditional hydrological models. Shorter time series do not provide enough training examples for the LSTM models to learn the patterns and relationships required to provide the desired hydrological simulations (Ayzel and Heistermann, 2021; Gauch et al., 2021b), whereas hydrological models can be fitted using relatively fewer observation data points (Perrin et al., 2007) as they already contain information in the form of the expected physics. However, the required length of data for proper training

might also depend on the number of contributing catchments in a large-sample dataset such as in this study (Gauch et al., 2021b) because the LSTM model training operates on pooled data from many catchments, increasing the sample sizes significantly. In this study, only catchments with long records of meteorologic and streamflow observations were used, which probably favors the LSTM models in all cases. In the case where only a few years of data per catchment were available, it is possible that the traditional hydrological models could outperform the LSTMs when modelling a single catchment. Of course,

in a large-sample application, the LSTM can then be trained using data from multiple catchments, so the amount of data per catchment that is required for LSTMs to outperform traditional hydrological models likely also depends on the number of catchments used in training. Also, it is important to note that hydrological models could, in theory, be regionalized even if they have different spatial discretization schemes. For example, a larger basin could be discretized and modelled in a semi-

distributed fashion, and parameters could still be regionalized. However, for an LSTM model, the size of the inputs must
remain the same throughout the training and regionalization process for all catchments. Therefore, it would not be possible to use a single weather station on a small catchment and then use a few weather stations on a larger catchment for LSTMs, since all catchments must share the same input format. Finally, another potential problem with LSTMs is that of structure design and hyperparameterization, as described in the next section.

## 5.2 LSTM structure design and hyperparameter selection

One of the unresolved problems in deep learning in general is that of neural network architecture design. Different types of model layers are available from the packages in commercial and open-source software and choosing which ones to implement for a given problem is not a trivial task. In the case of time-series simulations, LSTMs have shown to be excellent, but other RNN such as Gated Recurrent Units (GRU), which can be faster and use less memory than LSTM, have also been used in the literature to this end (Ayzel and Heistermann, 2021). Then, the user must decide the depth and scope of the model. Currently,
unless a user has a lot of experience with their data and neural network building, the best approach as recommended in the literature is to try multiple structures and optimize the hyperparameters to develop the best performing model (Jin et al., 2019). This means that many trials using various numbers of layers, and the complexity of each layer, must be performed, which can quickly become an intractable problem. Some software tools can help explore possible structures automatically, but the problem remains that the required complexity of such models depends on the data characteristics.

In this study, the largest models with 1x512 units, 2x256 units and 2x512 units did not perform statistically differently from one another, even though the 2-layer 512-unit model is much longer to train than the others. This might be because of limitations in the amount of provided data, or due to the structure not maximizing the information content. In any case, for this regionalization study, larger models did not seem to bring performance increases beyond 512 units, which is double the value
used in Kratzert et al. (2019b). However, an increase from 256 to 512 units did provide marginal performance gains, but these must be weighed against the associated increases in model training time. For this study, we opted to preserve the more complex model even though the gains were marginal due to not being limited by time or being in an operational context. In such cases, a simpler model (either 1x256 or 1x512 units) would have been perfectly acceptable. Alternatively, more computing power could be deployed to accelerate training in such cases. It is important to note that for this study, regionalization was performed
on 148 catchments, whereas in a real-world application, the target would most likely be a single catchment of interest, therefore reducing the computation effort by two orders of magnitude. Finally, the application of regularization on the bias, kernel and weights did not improve the testing skill in regionalization (results not shown). Regularization can sometimes help improve the model robustness by setting very small weights to zero, thus removing connections that could lead to overfitting. The remaining weights are therefore theoretically more likely to be those that represent the data structure and transformation to
obtain the streamflow. However, in this study, regularization failed to improve results. It is possible that this is due to the inherent uncertainty of streamflow making it difficult for the model to simulate the flows as if they were unbiased and error-

free at the pseudo-ungauged sites. Nonetheless, using dropout layers during training did manage to improve robustness by randomly dropping some neurons during training. While dropout rates were varied from 0.1 to 0.7, trial and error allowed finding a robust dropout layer of 0.3 for the LSTM layers and 0.1 for the dense layers which offered the best performance in
this regionalization study.

## 6 Conclusion

This study revisited past research on streamflow prediction in ungauged basins by comparing classical regionalization methods to the state-of-the-art LSTM deep learning model. Three hydrological models using widely-used regionalization approaches were tested on 148 catchments in northeast North America, and their results were compared to a simple LSTM model. The
results in this study showed that the LSTM model generally outperformed the hydrological model-based methods using the same available data. Furthermore, multiple LSTM hyperparameterizations showed the same improvements over the hydrological models, which attests to the LSTMs capacity to infer relationships between meteorologic data, catchment descriptors and streamflow, even without any explicit knowledge on hydrological processes.

The catchments in this study were all from the same region, but given the learning ability of the model, it should be possible to train models on large-sample hydrological datasets and feed more data to the models in order to maximize the ability to infer the hydrological processes by LSTMs and other recurrent neural networks. Kratzert et al. (2019b) already showed that this was possible over the continental United States, thus future work could continue in this direction and use larger and more diverse catchments across the world. Multiple such datasets already exist on continental or national scales, such as the HYSETS
database in North America (Arsenault et al., 2020), and the CAMELS datasets in the United States (Addor et al., 2017), Chile (Alvarez-Garreton et al., 2018), France (Delaigue et al., 2022), and UK (Coxon et al., 2020), which are prime candidates for training of regional hydrological models. Since deep learning models can make use of catchments with limited availability by pooling them with all the other available datasets, this makes LSTMs especially attractive for regionalization studies.

This study also showed that the LSTM model was able to provide streamflow time series at ungauged sites using relationships inferred from other sites, and that in many cases, the estimated streamflow was more accurate than that obtained from the hydrological models specifically calibrated on the data. The LSTM model therefore seems to have a better representation of the rainfall-runoff process as learned from the data directly, than the hydrological models have in their conceptualisation.

Given the previous studies in the literature as well as the resounding results obtained in this study, it is likely that the era of machine learning is here to stay in the field of streamflow prediction in ungauged basins. Hydrological models can still provide important details on the inner workings of the hydrological cycle in these types of studies, but if the only variable to predict is streamflow, then hydrological models are most likely not going to be able to contend as viable alternatives in the near future.

Future research should investigate the possibility of including larger datasets during training to improve the feature
representation and robustness across varying hydroclimatological conditions.

## 7 Code and data availability statement

All hydrometeorological and catchment descriptor data for this project were taken from the HYSETS dataset available at
https://osf.io/rpc3w/. The extracted and processed data for the 148 basins as well as the LSTM model codes are available at:
https://osf.io/3s2pq/.

## 8 Author contribution

RA, JLM and JM designed the experiments, and RA, JLM and F.Brunet performed them. RA, JLM, F.Brissette and JM
analyzed and interpreted the results. RA wrote the manuscript with significant contributions from JLM and F.Brissette. JM
and F.Brunet also provided editorial comments on initial drafts of the paper.

## 9 Competing interests

The authors declare that they have no conflict of interest.

## 10 Acknowledgements

This study was partly funded by the Natural Sciences and Engineering Research Council of Canada (NSERC), grant number
RGPIN 2018-04872. The authors would like to thank the two anonymous reviewers whose comments helped shape the paper
into its current form.

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

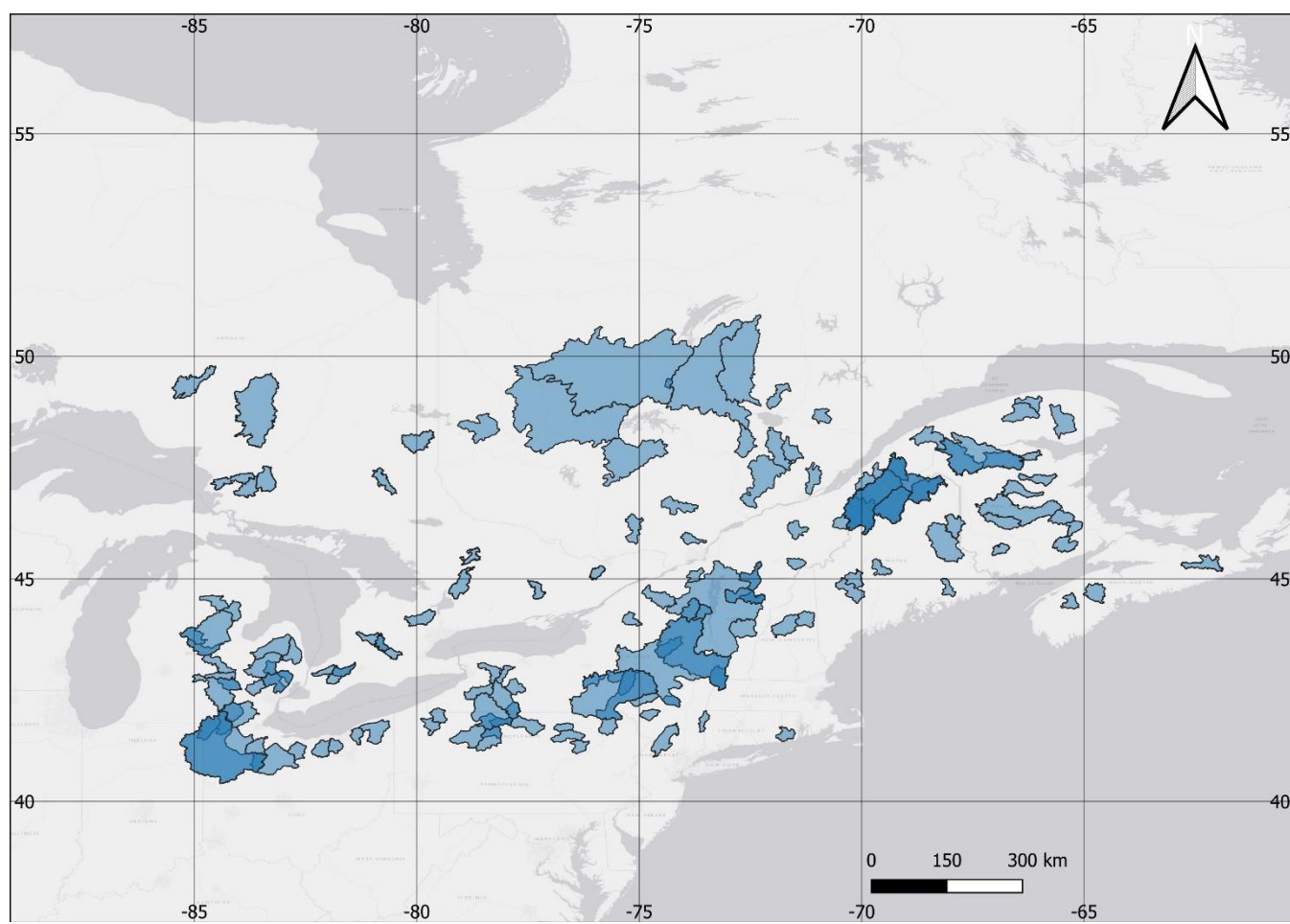

**Figure 1: Study site of the 148 catchments in Northeast North America. All catchments have the same transparent color, and regions with darker color (blue) represent areas where multiple catchments overlap.**

805

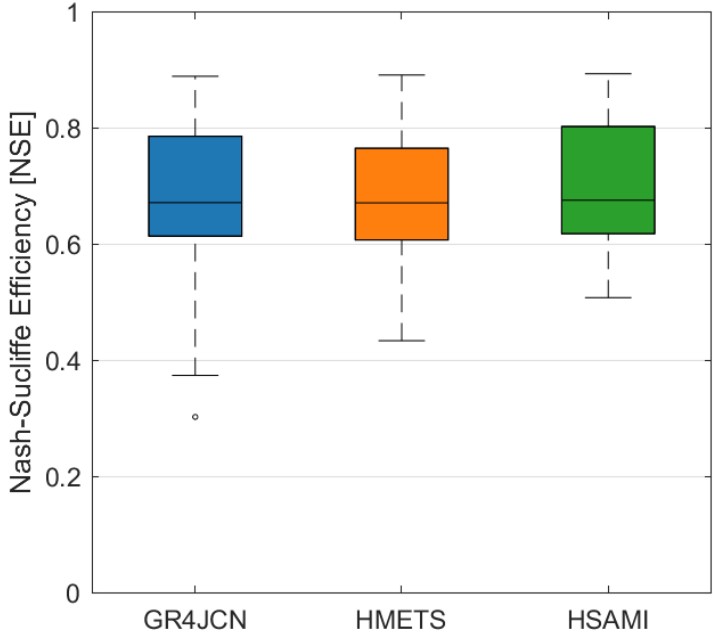

**Figure 2: Boxplots of Nash-Sutcliffe Efficiency (NSE) calibration scores for GR4JCN (blue), HMETS (orange) and HSAMI (green). The box represents the interquartile range (25th to 75th percentiles) with the median displayed as the horizontal line in the box. The whiskers represent the maximum and minimum non-outlier values, which are set at 1.5 times the interquartile range covering 99.3% of the distribution. Each boxplot contains the score for the 148 catchments calibrated on all data available between 1979 and 2018, using the first available year for each catchment as a warmup period.**

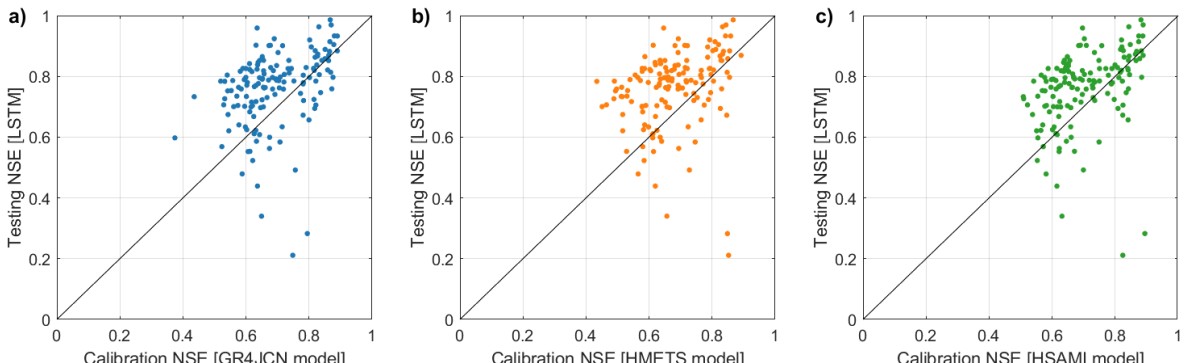

**Figure 3: Performance of hydrological models calibrated at each of the 148 study basins individually against the performance of the LSTM model in leave-one-out cross-validation (LOOCV) where the ungauged basin in question is not included in the set of basins used to train the LSTM and where the LSTM is trained and validated on 80% and 20% of the gauged basins, respectively. The hydrological models' performance displayed is hence a calibration performance while the LSTM performance is the testing performance (on ungauged locations). The models are evaluated using all available streamflow data in the period 1979 to 2018. The results are compared between LSTM models (y-axis) and three hydrological models (x-axis), i.e., a) GR4JCN, b) HMETS and c) HSAMI.**

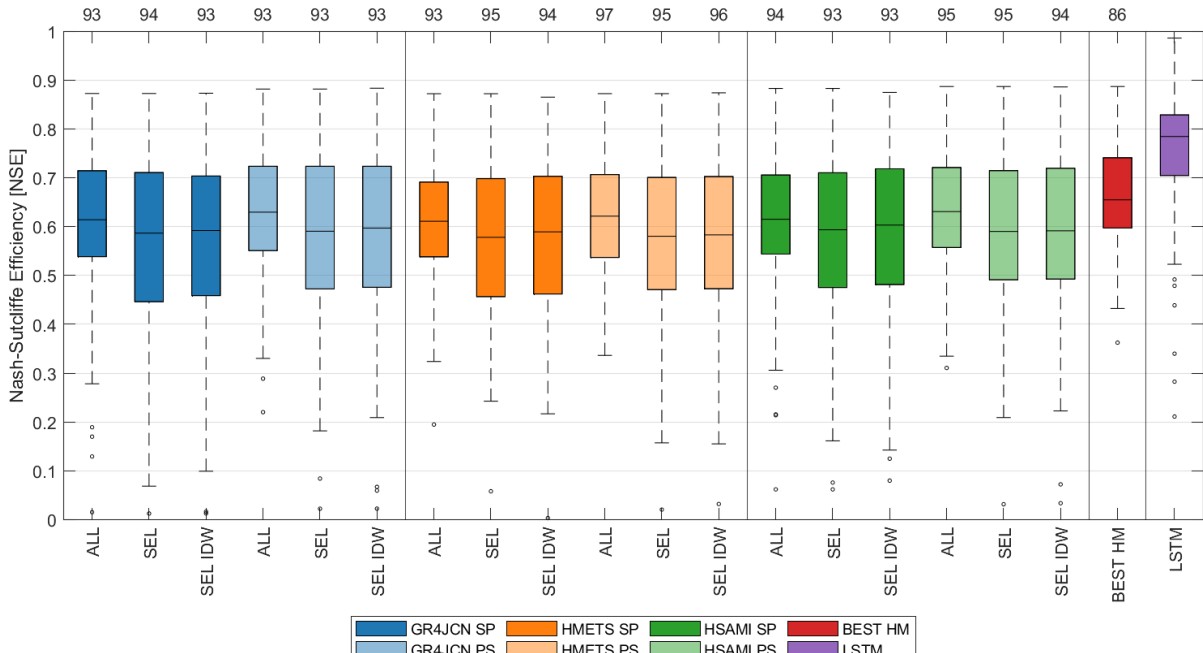

**Figure 4: Boxplots of the regionalization results for GR4JCN (blue), HMETS (orange) and HSAMI (green) using the spatial proximity (SP; dark) and physical similarity (PS; light) regionalization methods on all (ALL) or a selection (SEL) of catchments based on performance and application of inverse distance weighting (IDW) on the 148 catchments. All hydrological model-based regionalization methods implemented a multi-donor approach using five donors. Two additional boxplots present the best value from all hydrological model approaches (out of 18) for each catchment (BEST HM; red) and for the LSTM model (purple). The results displayed for the LSTM are the same results as they were used in Fig. 3. The models are evaluated on all available streamflow data available during the period 1979 to 2018. The box of each boxplot indicates the 25th and 75th percentile; the center line is the median; The whiskers represent the maximum and minimum non-outlier values, which are set at 1.5 times the interquartile range and covers 99.3% of the distribution. Values above each boxplot represent the percentage of basins for which the LSTM model performs better than the hydrology-model based regionalization method.**

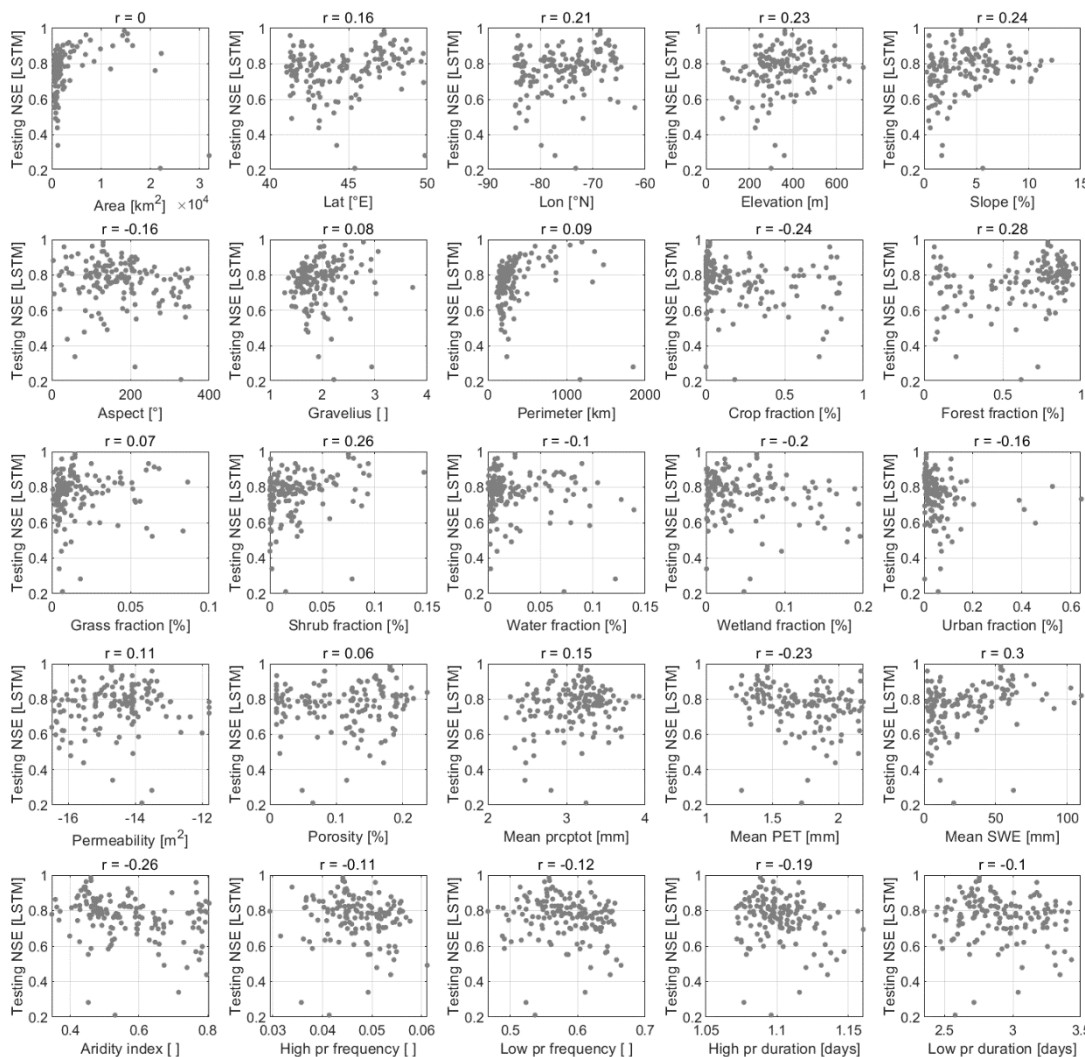

**Figure 5: Scatter between LSTM testing NSE and the various catchment descriptors used in this study (Table 1) for each of the 148 catchments studied. Correlation coefficients *r* are displayed in the title of the figure.**

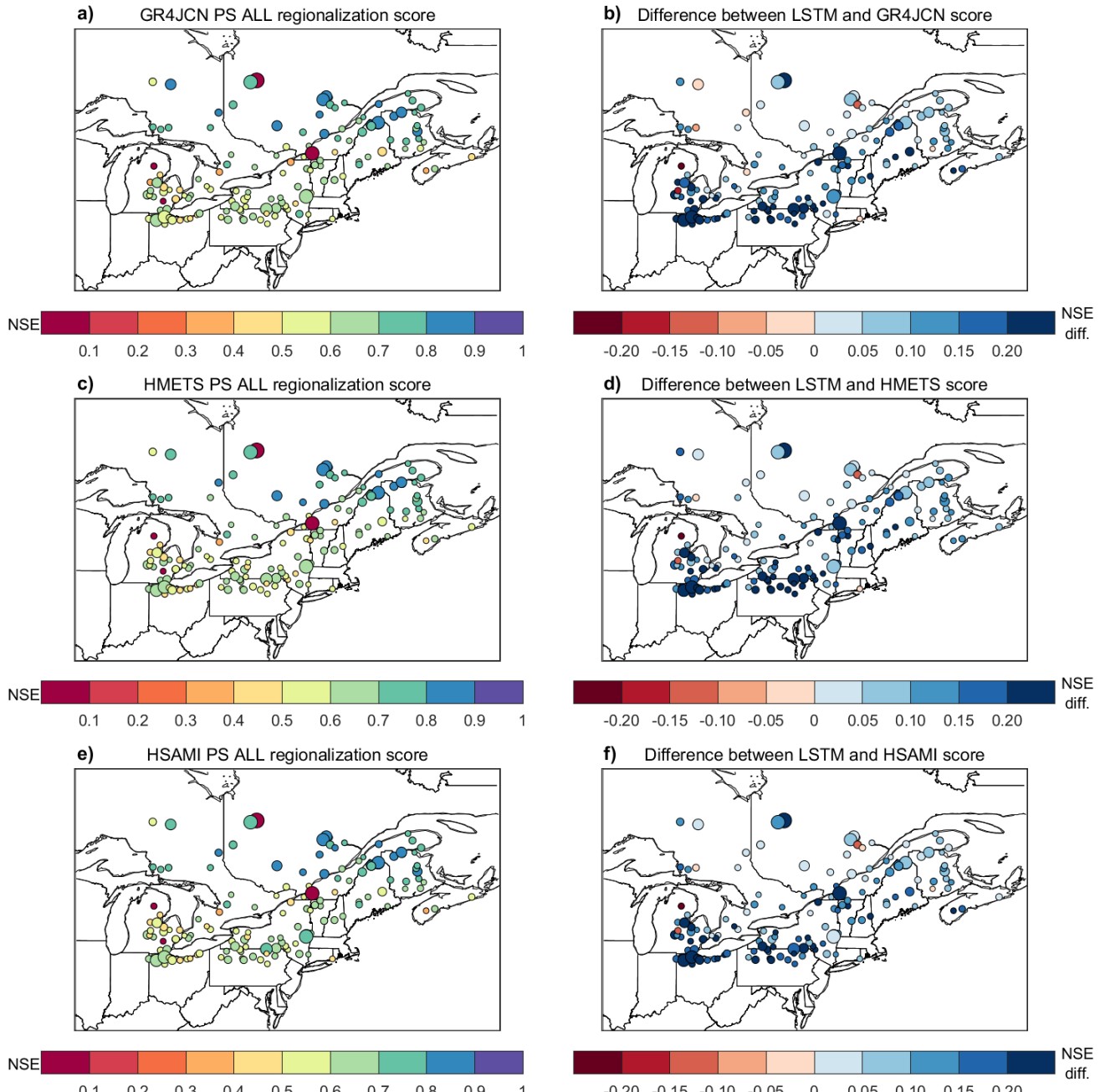

**Figure 6: Spatial distribution of regionalization skill of the hydrological models (a, c, e) and difference between the LSTM testing and hydrological model regionalization skill (b, d, f) for the three models GR4JCN (a, b), HMETS (c, d) and HSAMI (e, f). Circle sizes represent the relative sizes of the catchments and are placed at the catchment centers. The regionalization method with the best overall median for each hydrological model was selected for the comparison (here PS ALL for all three models). Blue colors in panels (b, d, f) indicate the LSTM is outperforming the hydrological model, while red colors indicate the opposite.**

840

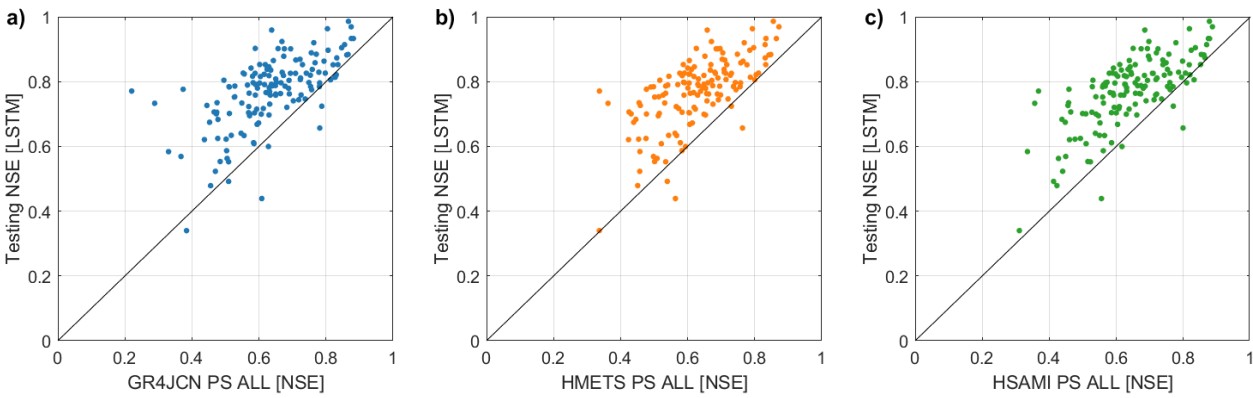

**Figure 7: Hydrological model regionalization NSE compared against the LSTM model in testing for a) GR4JCN, b), HMETS and c) HSAMI. The regionalization method with the best overall median for each hydrological model was selected for the comparison (here PS ALL for all three models). One point per catchment is displayed in each panel, for a total of 148 points.**

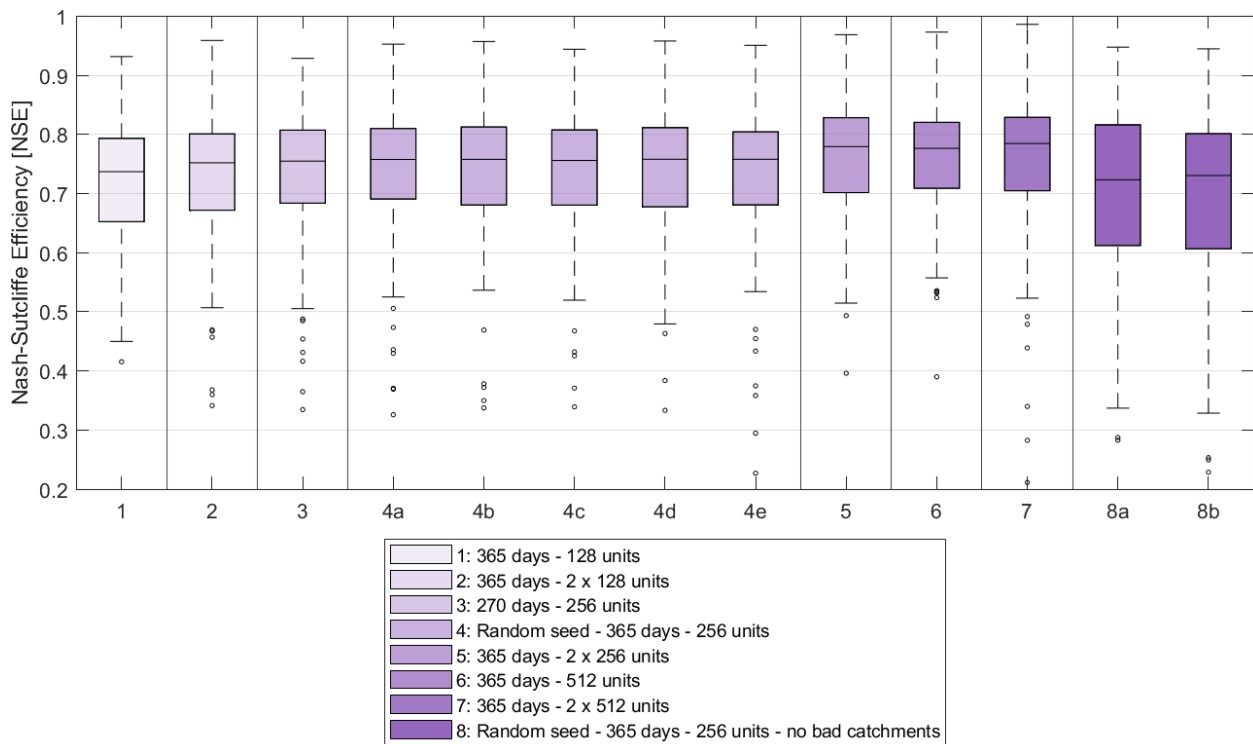

**Figure 8: Distribution of LSTM performance in leave-one-out cross-validation (LOOCV) over the 148 basins using different hyperparameters. The distribution used in this study is model #7. Details about the hyperparameter variations can be found in Table 2.**