# Peer review of "Continuous streamflow prediction in ungauged basins: Long Short-Term Memory Neural Networks clearly outperform traditional hydrological models"

_Hydrology and Earth System Sciences, 2022_

## Referee Comment (RC1)

Review of

Continuous streamflow prediction in ungauged basins: Long Short-Term Memory Neural Networks clearly outperform hydrological models

by Richard Arsenault, Jean-Luc Martel, Frédéric Brunet,
François Brissette, and Juliane Mai
[doi.org/10.5194/hess-2022-295]

General evaluation

This paper deals with a highly current subject matter. It presents a methodological framework for streamflow prediction in ungauged basins using a leave one out cross-validation approach (LOOCV) for three hydrological models and an LSTM network. The authors compare the performance of these four models for 148 basins in Northeast North America. The evaluation of the models at such a scale and working with these number of watersheds is quite impressive. The paper is well written, illustrated and organized. However, some points need to be taken into consideration to improve the manuscript, namely:

(i)     A more comprehensive literature review should be provided; that is, the authors should acknowledge the works of several other authors who have dealt with streamflow prediction using LSTMs, not just focusing on the works of a specific research group. Please homogenize the diversity of the literature review and cover the works of others that have provided significant achievements in the field of hydrological modelling using Deep Learning (DL) models.

(ii)    In the proposed LOOCV for LSTM modeling, the LSTM model was trained using a large dataset (N-1 basins), while keeping one basin as a pseudo-ungauged basin for validation. This approach departs from the basic philosophy of training DL models. Indeed, to avoid introducing a bias during training of DL models such as LSTM, overfitting should be avoided by considering a considerable proportion of the whole dataset as a testing dataset. What was the rationale behind this methodological approach?

(iii)   The authors propose an LSTM modeling approach for ungauged basins that will, without a doubt, spur the interest of the readers. However, the literature has provided several good performances of LSTM models for similar regions in Northeastern North America. Perhaps the authors could provide some insights for future work in dry regions where the presence of extreme flows may not be as prevalent and whether they expect that there approach would need to be modified or not accordingly.

At this point, I am looking forward to reading the authors' point of view as I believe they have earned an opportunity to provide sound rebuttal comments as I feel the paper has the potential to be a valuable contribution to *Hydrology and Earth System Sciences*. Thus, for the time being, I would say that major revisions are necessary and required.

Please find additional suggestions/recommendations and editorial comments below that will need to be addressed thoroughly before the paper can be recommended for publication.

Comments/suggestions/recommendations

| | |
|---|---|
| P4 | The following sentence, « In the Kratzert *et al.* (2018) study, the regional LSTM models performed on average just as well as the local LSTM with the median NSE difference of 0. » Local LSTM should be clarified compared to regional LSTM. |
| P5 | As illustrated in Figure 1 and Table 1, very large basins are included in the dataset, while including these basins during LSTM modeling has been quite a challenge since the input data are at the basin scale. How do the authors evaluate their results by assigning just one point to a basin with an average area of almost 31,900 km$^2$? |
| P10 | Why did the authors choose the leaky ReLU activation function? The authors should provide a table presenting the tested functions and values of the specificities of the LSTM model and the optimal ones; that would provide more insights to the readers. |
| P11 | Correct me if I am wrong, but according to the following sentence: « The twelve static descriptors presented in Table 1 allow the model to distinguish between each catchment ». Which one of them did the authors exactly use? Please provide another table introducing the list of twelve basin descriptors used for LSTM modeling. |
| P11 | According to the following sentence: « Static descriptors were normalized between 0 and 1 using a min-max scaler, while the dynamic variables were standardized by the mean and the standard deviation, which is a standard practice ». Did the authors include streamflow (target) during this normalization process? If not, how do they analyze their results after denormalization? Later, on the same page, it is mentioned, « The specific streamflow was used as the target variable by diving streamflow records by the drainage area, then converted from m$^3$s$^{-1}$ to mm.d$^{-1}$.». Please further clarify. |
| P14 | According to the following sentence: « This is important, considering that a strong performing hydrological model with the |

best regularization method is still outperformed on average by a relatively simple LSTM model. », the authors claim to use a simple LSTM model while using 2 LSTM layers each with 512 units, based on my experience, this is not considered a simple LSTM model. Please modify the text accordingly.

P14        Please be specific. According to the following sentence: « It is also important to note that the training (80%) and validation (20%) basins are categorized as such randomly, so the training step is performed on different catchments for each of the 5 runs #4a-#4e. », the authors should provide more details on how they couple this splitting approach with LOOCV, this needs to be clarified.

P14        Figure 8 shows the sensitivity of the hyperparameter selection and the assessment of the LSTM model structure. The authors claimed that the performance generally increases with a more complex model structure, meanwhile Figure 8 shows that increments are very minor between the simple structure models and the complicated models. In real-world practices, training and calibration of complex models face major challenges, how do the authors explain the choice of the selected complex model?

P15        According to the following sentence: « First, the nature of the LSTM model makes it extremely difficult or practically impossible to determine the logical flow of data between the observations and the predicted streamflow », readers may find it misleading since understanding the relationships between inputs and output of data-driven models can be achieved using sensitivity analysis. It is the authors' responsibility to provide such analysis as it would provide a way of following the logical flow of data. Thus, this sentence should be clarified accordingly.

P16        Based on the following sentence: « However, in this study, regularization failed to improve results ». Did the authors test all the possible values of dropout rates to reach such a conclusion? For instance, the value of 0.5 for the dropout rate has shown to be promising in improving the accuracy of streamflow modeling in other studies. Did the authors test this value?

Figures and Tables

None, all the tables and figures are well organized.

Editorial comments

None, this is a well-written paper.

---

## Author Comment (AC3)

**Reviewer #2**

General comments

This manuscript is a positive addition to the growing amount of research on the use of machine learning techniques in hydrological modelling with a focus on ungauged basins. The study compares an LSTM-based model trained over multiple catchments with three traditional hydrological models calibrated using several regionalization methods. Overall, the LSTM outperformed the traditional hydrological models at almost all catchments regardless of regionalization method used. This manuscript provides interesting results, is well structured, and was enjoyable to read. However, some additional clarifications throughout the manuscript would allow the reader to fully understand the chosen methodology and the presented results. Please see the specific comments below.

We would like to thank Reviewer #2 for their positive comments and for the suggestions on how to improve the manuscript. We have provided a point-by-point reply to all issues and comments below.

Specific comments

Introduction: As the authors rightly point out the LSTM has been used in several studies in recent years. However, the literature review is mainly focused on work conducted on catchments in North America and with limited acknowledgment of studies conducted in other regions (e.g., Choi et al., 2022; Nogueira Filho et al., 2022; Ayzel et al., 2021; Ayzel et al., 2020). Additionally, it would be beneficial to include a couple of lines near the beginning explaining that this study uses regionalization of hydrological model parameters specifically, and briefly defining what is meant by "hydrological model".

This was also highlighted by Reviewer #1, and we will indeed improve the literature review. We will expand it to a more global scale and will also clearly define our interpretation of "hydrological model" and regionalization. Citations that will be added include the following:

Ayzel G, Kurochkina L, Abramov D, Zhuravlev S. Development of a Regional Gridded Runoff Dataset Using Long Short-Term Memory (LSTM) Networks. Hydrology. 2021; 8(1):6. https://doi.org/10.3390/hydrology8010006

Ayzel, G., Kurochkina, L., Kazakov, E., & Zhuravlev, S. (2020). Streamflow prediction in ungauged basins: benchmarking the efficiency of deep learning. In E3S Web of Conferences (Vol. 163, p. 01001). EDP Sciences.

Choi, J., Lee, J., & Kim, S. (2022). Utilization of the Long Short-Term Memory network for predicting streamflow in ungauged basins in Korea. Ecological Engineering, 182, 106699.

Li, X., Khandelwal, A., Jia, X., Cutler, K., Ghosh, R., Renganathan, A., Xu, S., Tayal, K., Nieber, J., Duffy, C. and Steinbach, M., 2022. Regionalization in a global hydrologic deep learning model: from physical descriptors to random vectors. Water Resources Research, 58(8), p.e2021WR031794.

Nogueira Filho, F. J. M., Souza Filho, F. D. A., Porto, V. C., Vieira Rocha, R., Sousa Estácio, Á. B., & Martins, E. S. P. R. (2022). Deep Learning for Streamflow Regionalization for Ungauged Basins: Application of Long-Short-Term-Memory Cells in Semiarid Regions. Water, 14(9), 1318.

Line 210-212: I am confused by the sentence "Each of these models was calibrated using the Covariance Matrix Adaptation Evolution Strategy (CMAES; Hansen et al., 2003) optimization algorithm in the Arsenault and Brissette (2014) study, and parameters are reused here to maintain the comparability to this study." Was the HSAMI model not the only hydrological model used in the Arsenault and Brissette (2014)? Please clarify which parameters are reused, and how they relate to the calibration method and results described in lines 210-225.

This is a good catch. The models were actually used across a few papers instead of only the one mentioned previously:

Arsenault and Brissette 2014: HSAMI

Arsenault and Brissette 2015: HSAMI, HMETS

Poissant et al. 2017: GR4JCN

The information of the parameters that was taken from the other studies were the parameter boundaries for calibrations and not the calibrated parameters themselves since the models were applied to different catchments. This will be clarified in the text, referring to each study for each model and providing the clarifications regarding model calibration.

Line 262: Why was N=5 chosen (over other values between 4-8)? Please state the reasoning.

The value of N=5 was chosen purely due to it being recommended in the literature as a reasonable value in the 4-8 range. In many papers, values between 4-8 donors do not show any significant differences, and between 5-7 previous studies have shown that there is essentially no difference. So, N=5 was chosen to make sure the full effect of multi-donor averaging was at play while not using unnecessary computing resources to extend to 6, 7 or 8 donors. This will be clarified in the text.

Line 275: Please state how many catchments were classified as "poor" and thus removed when the filter was applied.

This will be added in the text. As seen in Figure 2, slightly more than half of the catchments have calibration NSE values below 0.7. This ranges from 84 to 89 basins depending on the hydrological model.

Line 307-308: "The twelve static descriptors presented in Table 1 allow the model to distinguish between each catchment.". Highlighting these variables in Table 1 may make it easier to understand which 12 are used as input to the LSTM. Also land cover (%) is split into 7 entries in Table 1 but I think is only considered as 1 of the 12 static descriptors which is confusing.

Good catch, and we will copy a response given to Reviewer #1 to this effect for coherence and for your convenience:

This is an error related to the fact that in our first simulations we were using 12 descriptors, until we found that using more (many from recommendations in the literature) allowed for better results. We redid all the simulations but forgot to update this part of the text. It will be corrected in the next version. All 25 catchment descriptors listed in Table 1 were used in this study.

Also, the 7 land cover classes are all considered independently and count towards 7 of the 25 descriptors used.

Line 312: Please clearly define the training, validation, and testing catchments.

We have answered a similar question from Reviewer #1, and the response has been copied here for your convenience:

Indeed, this will be clarified along with the general comment #2 above (from Reviewer #1) regarding the training/validation/testing phases. Essentially, every time a model is trained, 1 catchment (pre-determined as the pseudo-ungauged basin) is removed from the lot. Then, remaining basins are split into 2 groups, i.e., training (80%) and validation (20%). This splitting is random in nature. While the testing dataset could have been chosen as a fixed percentage (e.g., 20%) of all watersheds, using a leave-one-out-cross-validation (LOOCV) methodology was essential to compare results to previous studies.

Line 330: Why was model #7 chosen as the LSTM structure of choice? Please state the reasoning.

This point was also raised by Reviewer #1, and we have provided the following response, for your convenience:

In figure 8, we can see that the trend is monotonously increasing from model 1 to 7, in increasing complexity order. The median testing NSE increases from 0.74 for the simple model to 0.785 for the more complex model. Furthermore, each of the quantiles of the distributions are improving with each successive model. These types of improvements in regionalization are very significant. Therefore, the most complex model was selected since it outperformed the others, without requiring the modeller to integrate new physics/physical process representation. Simply by adding LSTM layers, the LSTM model was able to perform better in testing/regionalization mode, at the expense of computing time. […] Therefore, if the computing time is available, the more complex model is to be preferred. Especially since the training is only performed once for a given ungauged catchment application. A section to this effect will be added to the text, detailing this choice.

Line 374-375: Were non-linear relationships between catchment descriptors and NSE values considered?

At this stage, no, only linear relationships were considered, to see if there was a correlation (i.e., if perhaps larger basins reacted better than smaller basins, etc.). However, this was not the case, leading to believe that the LSTM was able to use these descriptors in a non-linear fashion to provide the good, basin-dependent regionalized streamflow.

Line 395-396: "relatively simple LSTM model". Is this still referring to model #7 which is the most complex of the LSTM models tested? Please clarify. Also, on line 489 - "simple LSTM model".

Thank you for this comment. Again, we refer to a response given to Reviewer #1 to this same question for your convenience:

The interpretation is correct, the text does mention that it is a "simple LSTM model". However, this is relative, as in our opinion, an LSTM (even if the structure is internally quite complex) did not require much setting-up, calibrating, adjusting, etc. compared to other, more classical hydrological models. It is true that the LSTM model structure is quite complex compared to others, so this text will be modified to reflect this, i.e., that the LSTM model is complex but can be applied without a lot of work to represent the specific processes etc.

In the text, we will also clarify the meaning of "complexity" in the context of our study.

Lines 455-459: As discussed in the introduction (lines 119-126) traditional hydrological models and LSTM models show different behaviours in terms of performance for increasing lengths of data (e.g., the plateauing after 3 years of the GR4J model (line 122)). Please comment on the "fair-ness" of the comparison considering only catchments with at least 30 years of data are included?

This is a fair point, and we will add a discussion point to reflect on it. In theory, both he GR4JCN and the LSTM model have access to the same data and as such, the comparison is as fair as it can be. However, GR4JCN must compromise on the parameter sets to use to be "generally" good, whereas the LSTM has many more degrees of freedom to fit to various hydroclimatological situations. However, GR4JCN has a predefined structure where processes are directly defined, whereas the LSTM must build its internal structure using its more numerous "parameters". We will add a discussion point detailing the fact that not only the LSTM is to be favored due to the long time series of available data, but also that it can ingest data from many more catchments as well, whereas GR4J is limited to containing information from one catchment at a time.

Technical corrections

These technical corrections will also all be addressed in the revised version of the manuscript:

Line 12: Suggest changing "A series of …" to "a set of …" as series implies that there is a sequential element to the methods.

Line 12: "regionalization methods are applied"

Line 180-181: "Environment and Climate Change Canada (ECCC), and the United States Geological Survey (USGS)."

Line 232: Suggest changing "for each scenario" to "for each of the 18 scenarios" for clarity.

Line 288: "have difficulty remembering"

Line 315: "then converted from m.s-1 to mm.d-1" (as the division by drainage area would already have removed two spatial dimensions).

References

Choi, J., Lee, J., & Kim, S. (2022). Utilization of the Long Short-Term Memory network for predicting streamflow in ungauged basins in Korea. Ecological Engineering, 182, 106699.

Nogueira Filho, F. J. M., Souza Filho, F. D. A., Porto, V. C., Vieira Rocha, R., Sousa Estácio, Á. B., & Martins, E. S. P. R. (2022). Deep Learning for Streamflow Regionalization for Ungauged Basins: Application of Long-Short-Term-Memory Cells in Semiarid Regions. Water, 14(9), 1318.

Ayzel G, Kurochkina L, Abramov D, Zhuravlev S. Development of a Regional Gridded Runoff Dataset Using Long Short-Term Memory (LSTM) Networks. Hydrology. 2021; 8(1):6. https://doi.org/10.3390/hydrology8010006

Ayzel, G., Kurochkina, L., Kazakov, E., & Zhuravlev, S. (2020). Streamflow prediction in ungauged basins: benchmarking the efficiency of deep learning. In E3S Web of Conferences (Vol. 163, p. 01001). EDP Sciences.

---

## Author Response (AR1)

We would first like to thank the Reviewers, Editor and community members for their comments and suggestions to help improve this paper. This document presents the modifications made to the original manuscript in response to the reviewer comments and suggestions. Original comments are in black font and author responses are in blue.

**Reviewer #1**

This paper deals with a highly current subject matter. It presents a methodological framework for streamflow prediction in ungauged basins using a leave one out cross-validation approach (LOOCV) for three hydrological models and an LSTM network. The authors compare the performance of these four models for 148 basins in Northeast North America. The evaluation of the models at such a scale and working with these number of watersheds is quite impressive. The paper is well written, illustrated and organized. However, some points need to be taken into consideration to improve the manuscript, namely:

We would like to thank Reviewer #1 for their comments and suggestions. Here are the point-by-point responses on how the manuscript was revised.

(i)   A more comprehensive literature review should be provided; that is, the authors should acknowledge the works of several other authors who have dealt with streamflow prediction using LSTMs, not just focusing on the works of a specific research group. Please homogenize the diversity of the literature review and cover the works of others that have provided significant achievements in the field of hydrological modelling using Deep Learning (DL) models.
We have explored other studies and applications and contextualized the work better in this regard. We have found relevant studies performing similar work in different regions of the world and have added a more in-depth context to the introduction. The following references were added to the paper:

Ayzel G, Kurochkina L, Abramov D, Zhuravlev S. Development of a Regional Gridded Runoff Dataset Using Long Short-Term Memory (LSTM) Networks. Hydrology. 2021; 8(1):6. https://doi.org/10.3390/hydrology8010006

Ayzel, G., Kurochkina, L., Kazakov, E., & Zhuravlev, S. (2020). Streamflow prediction in ungauged basins: benchmarking the efficiency of deep learning. In E3S Web of Conferences (Vol. 163, p. 01001). EDP Sciences.

Choi, J., Lee, J., & Kim, S. (2022). Utilization of the Long Short-Term Memory network for predicting streamflow in ungauged basins in Korea. Ecological Engineering, 182, 106699.

Li, X., Khandelwal, A., Jia, X., Cutler, K., Ghosh, R., Renganathan, A., Xu, S., Tayal, K., Nieber, J., Duffy, C. and Steinbach, M., 2022. Regionalization in a global hydrologic deep learning model: from physical descriptors to random vectors. Water Resources Research, 58(8), p.e2021WR031794.

Nogueira Filho, F. J. M., Souza Filho, F. D. A., Porto, V. C., Vieira Rocha, R., Sousa Estácio, Á. B., & Martins, E. S. P. R. (2022). Deep Learning for Streamflow Regionalization for Ungauged Basins: Application of Long-Short-Term-Memory Cells in Semiarid Regions. Water, 14(9), 1318.

Zhang, Y., Ragettli, S., Molnar, P., Fink, O. and Peleg, N., 2022. Generalization of an Encoder-Decoder LSTM model for flood prediction in ungauged catchments. Journal of Hydrology, p.128577.

(ii)    In the proposed LOOCV for LSTM modeling, the LSTM model was trained using a large dataset (N-1 basins), while keeping one basin as a pseudo-ungauged basin for validation. This approach departs from the basic philosophy of training DL models. Indeed, to avoid introducing a bias during training of DL models such as LSTM, overfitting should be avoided by considering a considerable proportion of the whole dataset as a testing dataset. What was the rationale behind this methodological approach?

This was incorrect and we acknowledged where the confusion came from. We forgot to clearly state the percentage of catchments used in training and validation in the methods and only referred to the values in line 408 of the original manuscript, where we explained that it is 80% of (N-1) basins used in training and 20% used in validation, with the pseudo-ungauged basin being the only one used as the "testing" basin. This is to ensure the LSTM model does not over-train/overfit, and the "regionalization" skill is evaluated on the completely independent testing basin. This has been clarified in lines 344-352 in the revised manuscript.

(iii)   The authors propose an LSTM modeling approach for ungauged basins that will, without a doubt, spur the interest of the readers. However, the literature has provided several good performances of LSTM models for similar regions in Northeastern North America. Perhaps the authors could provide some insights for future work in dry regions where the presence of extreme flows may not be as prevalent and whether they expect that there approach would need to be modified or not accordingly.

Thank you for this suggestion. We added a few sentences on the robustness of the approach on other regions in the introduction, lines 116-123. We did not perform new simulations and tests but a very recent study (Nogueira Filho et al. 2022) has shown that LSTM and another neural network were able to perform better than conceptual models in a semi-arid region of Brazil. It is thus likely that the LSTM, given sufficient data and trained on data from similar conditions, would outperform hydrological models in those regions as well, although this hypothesis would need to be tested in future studies.

At this point, I am looking forward to reading the authors' point of view as I believe they have earned an opportunity to provide sound rebuttal comments as I feel the paper has the potential to be a valuable contribution to Hydrology and Earth System Sciences. Thus, for the time being, I would say that major revisions are necessary and required.

Please find additional suggestions/recommendations and editorial comments below that will need to be addressed thoroughly before the paper can be recommended for publication.

We thank Reviewer #1 for these comments and will now respond to the specific comments below.

Comments/suggestions/recommendations

P4 The following sentence, « In the Kratzert et al. (2018) study, the regional LSTM models performed on average just as well as the local LSTM with the median NSE difference of 0. » Local LSTM should be clarified compared to regional LSTM.

This has been clarified as :

"In the Kratzert et al. (2018) study, the regional LSTM models (single models that can predict streamflow on a variety of catchments in a region) performed on average just as well as the local LSTM (trained specifically on a single catchment at a time) with a median NSE difference of 0." in lines 103-106.

P5 As illustrated in Figure 1 and Table 1, very large basins are included in the dataset, while including these basins during LSTM modeling has been quite a challenge since the input data are at the basin scale. How do the authors evaluate their results by assigning just one point to a basin with an average area of almost 31,900 km2?

The issue of scale is actually quite interesting and one that we detailed more in lines 470-475 of the revised manuscript. In a nutshell, for the "traditional" hydrological models, having a wide variety of catchments might actually be causing problems since they will be less similar to their smaller counterparts. It is probable that the hydrological models calibrated on the large catchments will not transfer very well to the smaller catchments due to differences in hydrological response. However, LSTM networks can make use of this information to build relationships using more diversity, allowing to detect patterns more clearly. Therefore, adding these larger catchments probably helps predicting flows on other catchments since it contributes extra data points in the model that will help avoid extrapolating during testing. As for the data being averaged at the catchment scale, this is indeed a limitation of lumped hydrological models, and feeding the same data to the LSTM seems to benefit the LSTM. One other limitation is that the LSTM applied in regionalization needs to have the same number of inputs as all the training sets, thus having variable numbers of inputs (e.g., meteorological stations) at each catchment would not be possible without using a much more complex LSTM structure. This was also detailed in lines 502-511 of the revised manuscript.

P10 Why did the authors choose the leaky ReLU activation function? The authors should provide a table presenting the tested functions and values of the specificities of the LSTM model and the optimal ones; that would provide more insights to the readers.

Most parameters were adjusted using expert knowledge to focus on the hyperparameters that had a good likelihood of returning good results. Also, the model structure was generally made to be similar to that of Kratzert et al. (2018) given their excellent results. Therefore only a few hyperparameters were adjusted, as displayed in table 2. As for the LeakyReLU activation function, it was used to eliminate any possibility of generating impossible objective function values or exploding gradients. This was not common with ReLU but depending on the objective function choice and other hyperparameters, we did encounter cases when the model would not converge or would return undefined objective function values. LeakyReLU minimizes these errors, at the expense of a bit more computing time. This has been revised in lines 316-319 of the revised manuscript.

P11 Correct me if I am wrong, but according to the following sentence: « The twelve static descriptors presented in Table 1 allow the model to distinguish between each catchment ». Which one of them did the authors exactly use? Please provide another table introducing the list of twelve basin descriptors used for LSTM modeling.

This was an error related to the fact that in our first simulations we were using 12 descriptors, until we found that using more (many from recommendations in the literature) allowed for better results. We redid all the simulations but forgot to update this part of the text. It has been corrected in this revised version. All 25 catchment descriptors listed in Table 1 were used in this study.

P11 According to the following sentence: « Static descriptors were normalized between 0 and 1 using a min-max scaler, while the dynamic variables were standardized by the mean and the standard deviation, which is a standard practice ». Did the authors include streamflow (target) during this normalization process? If not, how do they analyze their results after denormalization? Later, on the same page, it is mentioned, « The specific streamflow was used as the target variable by diving streamflow records by the drainage area, then converted from m3 s-1 to mm.d-1 .». Please further clarify.

No, the target streamflow data was not scaled using a min-max or standard scaler. We added a few sentences in lines 335-342 to make this clearer in the revised manuscript. Streamflow is not an input to the models; it is only used to evaluate the model performance (as for traditional hydrologic models). Only the inputs to the model were scaled in this manner. We have emphasized this in the manuscript. Working with input variables on a similar scale allows the model to converge faster with the use of larger learning rates. However, normalizing the target variable is not needed to accelerate the training and is typically not performed. On the other hand, streamflow is highly dependent of the drainage area, generating larger volumes for the same precipitation. While this information is available within the static descriptor, we found that including that knowledge upfront ends up accelerating the training process significantly instead of letting the model search for this correlation. The drainage area static variable remains useful for the model considering that hydrological processes with differ between small and large watersheds. Thus, streamflow is standardized by the drainage area, such that units are $m^3s^{-1}km^{-2}$. Then, by adjusting the length units, we can obtain the units mm3.s-1. This provides values that are hard to interpret/debug, so they are multiplied by the time units such that we obtain mm.d-1 units. This is what the LSTM model tries to reproduce and is trained on. Finally, once the LSTM returns a series of outputs for the pseudo-ungauged site, the reverse calculations are performed to obtain the flowrate to be compared to the observations. The fact that flows are compared on a mm/d basis means that the output can be tailored to any catchment.

P14 According to the following sentence: « This is important, considering that a strong performing hydrological model with the 3 | 4 best regionalization method is still outperformed on average by a relatively simple LSTM model. », the authors claim to use a simple LSTM model while using 2 LSTM layers each with 512 units, based on my experience, this is not considered a simple LSTM model. Please modify the text accordingly.

This interpretation is correct, the text did indeed mention that it is a "simple LSTM model". However, this is relative, as in our opinion, an LSTM (even if the structure is internally quite complex) did not require much setting-up, calibrating, adjusting, etc. compared to other, more classical hydrological models. It is true that the LSTM model structure is quite complex compared to others, so this text has been modified to reflect this, i.e., that the LSTM model is complex but can be applied without a lot of work to represent the specific processes etc. The changes have been included in lines 430-432, and indirectly in lines 530-535.

P14 Please be specific. According to the following sentence: « It is also important to note that the training (80%) and validation (20%) basins are categorized as such randomly, so the training step is

performed on different catchments for each of the 5 runs #4a-#4e. », the authors should provide more details on how they couple this splitting approach with LOOCV, this needs to be clarified.

Indeed, this has been clarified along with the general comment #2 above regarding the training/validation/testing phases in lines 344-352. Essentially, every time a model is trained, 1 catchment (pre-determined as the pseudo-ungauged basin) is removed from the lot. Then, remaining basins are split into 2 groups, i.e., training (80%) and validation (20%). This splitting is random in nature, thus when testing over a large series of basins (each of the 148 basins considered pseudo-ungauged one at a time) and while varying the model structure, it is clear that the stochastic component could play a role in the results. However, given the large number of such simulations, the expected variance from one test to the other should be very small.

P14 Figure 8 shows the sensitivity of the hyperparameter selection and the assessment of the LSTM model structure. The authors claimed that the performance generally increases with a more complex model structure, meanwhile Figure 8 shows that increments are very minor between the simple structure models and the complicated models. In real-world practices, training and calibration of complex models face major challenges, how do the authors explain the choice of the selected complex model?

In figure 8, we can see that the trend is monotonously increasing from model 1 to 7, in increasing complexity order. The median testing NSE increases from 0.740 for the simple model to 0.785 for the most complex model. Furthermore, each of the quantiles of the distributions are improving with each successive model. These types of improvements in regionalization are very significant. Therefore, the most complex model was selected since it outperformed the others, without requiring the modeller to integrate new physics/physical process representation. Simply by adding LSTM layers, the LSTM model was able to perform better in testing/regionalization mode, at the expense of computing time. It is true that building and training complex hydrological models, in the classical sense, requires a lot of effort. In this paper, we show that the performance can be better than what is obtained with hydrological models, but with little modelling effort from the hydrologist. Therefore, if the computing time is available, the more complex model is to be preferred, especially since the training is only performed once for a given ungauged catchment application. A section to this effect has been added to the text in lines 530-535.

P15 According to the following sentence: « First, the nature of the LSTM model makes it extremely difficult or practically impossible to determine the logical flow of data between the observations and the predicted streamflow », readers may find it misleading since understanding the relationships between inputs and output of datadriven models can be achieved using sensitivity analysis. It is the authors' responsibility to provide such analysis as it would provide a way of following the logical flow of data. Thus, this sentence should be clarified accordingly.

Agreed, the sentence was not entirely clear. What was meant was that the physical representation of processes is lost in these deep learning models. How precipitation becomes streamflow is hard to track due to the numerous weights, non-linear functions and layers that add lags and biases at each step. Therefore, the best approach would be to evaluate the final trained weights and try to correlate them with expected hydrological variables, but this was not part of the scope of this study. Some papers have already started showing these links (as stated in the manuscript) but it remains that following a precipitation value in an LSTM and seeing how it affects the streamflow for the next 3 days is very much

convoluted compared to a classical hydrological model, where each process is explicitly defined. This has been rephrased in the revised version of the manuscript in lines 485-486.

P16 Based on the following sentence: « However, in this study, regularization failed to improve results ». Did the authors test all the possible values of dropout rates to reach such a conclusion? For instance, the value of 0.5 for the dropout rate has shown to be promising in improving the accuracy of streamflow modeling in other studies. Did the authors test this value?

Yes, dropout values of 0.1 to 0.7 were tested, and it was found that the values proposed here performed best (dropout of 0.3 for the LSTM layers and 0.1 for the dense layers). Dropouts simply drop some neurons during training to make the model more robust. On the other hand, regularization attempts to set some weights to 0 overall in the final model to remove noise from neurons that are weak (close to zero) to begin with, removing some influence from noise overfitting. This means stronger, more important weights remain, that are more robust to the signal. However, using regularization did not improve results in this study. In our case, perhaps the dropout rate was sufficient to provide this reliability during the training, or perhaps the fact that there were a lot of training samples meant that the model was able to converge on the signal without too much overfitting in the first place. This has been clarified in the revised manuscript in lines 541-544.

**Reviewer #2**

General comments

This manuscript is a positive addition to the growing amount of research on the use of machine learning techniques in hydrological modelling with a focus on ungauged basins. The study compares an LSTM-based model trained over multiple catchments with three traditional hydrological models calibrated using several regionalization methods. Overall, the LSTM outperformed the traditional hydrological models at almost all catchments regardless of regionalization method used. This manuscript provides interesting results, is well structured, and was enjoyable to read. However, some additional clarifications throughout the manuscript would allow the reader to fully understand the chosen methodology and the presented results. Please see the specific comments below.

We would like to thank Reviewer #2 for their positive comments and for the suggestions on how to improve the manuscript. We have provided a point-by-point reply to all issues and comments below.

Specific comments

Introduction: As the authors rightly point out the LSTM has been used in several studies in recent years. However, the literature review is mainly focused on work conducted on catchments in North America and with limited acknowledgment of studies conducted in other regions (e.g., Choi et al., 2022; Nogueira Filho et al., 2022; Ayzel et al., 2021; Ayzel et al., 2020). Additionally, it would be beneficial to include a couple of lines near the beginning explaining that this study uses regionalization of hydrological model parameters specifically, and briefly defining what is meant by "hydrological model".

This comment is similar to that of Reviewer #1 and thus the response is copied here for your convenience:

We have explored other studies and applications and contextualized the work better in this regard. We have found relevant studies performing similar work in different regions of the world and have added a more in-depth context to the introduction. The following references were added to the paper:

Ayzel G, Kurochkina L, Abramov D, Zhuravlev S. Development of a Regional Gridded Runoff Dataset Using Long Short-Term Memory (LSTM) Networks. Hydrology. 2021; 8(1):6. https://doi.org/10.3390/hydrology8010006

Ayzel, G., Kurochkina, L., Kazakov, E., & Zhuravlev, S. (2020). Streamflow prediction in ungauged basins: benchmarking the efficiency of deep learning. In E3S Web of Conferences (Vol. 163, p. 01001). EDP Sciences.

Choi, J., Lee, J., & Kim, S. (2022). Utilization of the Long Short-Term Memory network for predicting streamflow in ungauged basins in Korea. Ecological Engineering, 182, 106699.

Li, X., Khandelwal, A., Jia, X., Cutler, K., Ghosh, R., Renganathan, A., Xu, S., Tayal, K., Nieber, J., Duffy, C. and Steinbach, M., 2022. Regionalization in a global hydrologic deep learning model: from physical descriptors to random vectors. Water Resources Research, 58(8), p.e2021WR031794.

Nogueira Filho, F. J. M., Souza Filho, F. D. A., Porto, V. C., Vieira Rocha, R., Sousa Estácio, Á. B., & Martins, E. S. P. R. (2022). Deep Learning for Streamflow Regionalization for Ungauged Basins: Application of Long-Short-Term-Memory Cells in Semiarid Regions. Water, 14(9), 1318.

Zhang, Y., Ragettli, S., Molnar, P., Fink, O. and Peleg, N., 2022. Generalization of an Encoder-Decoder LSTM model for flood prediction in ungauged catchments. Journal of Hydrology, p.128577.

Also, clarifications of our interpretation of "hydrological model" and "regionalization" have been added in lines 45 and 29, respectively.

Line 210-212: I am confused by the sentence "Each of these models was calibrated using the Covariance Matrix Adaptation Evolution Strategy (CMAES; Hansen et al., 2003) optimization algorithm in the Arsenault and Brissette (2014) study, and parameters are reused here to maintain the comparability to this study." Was the HSAMI model not the only hydrological model used in the Arsenault and Brissette (2014)? Please clarify which parameters are reused, and how they relate to the calibration method and results described in lines 210-225.

This is a good catch. The models were actually used across a few papers instead of only the one mentioned previously, and this has been corrected and clarified in lines 225-226 of the revised manuscript.

The information of the parameters that was taken from the other studies were the parameter boundaries for calibrations and not the calibrated parameters themselves since the models were applied to different catchments. This was clarified in the text in line 227.

Line 262: Why was N=5 chosen (over other values between 4-8)? Please state the reasoning.

The value of N=5 was chosen purely due to it being recommended in the literature as a reasonable value in the 4-8 range. In many papers, values between 4-8 donors do not show any significant differences, and between 5-7 previous studies have shown that there is essentially no difference. So, N=5 was chosen to make sure the full effect of multi-donor averaging was at play while not using unnecessary computing resources to extend to 6, 7 or 8 donors. This has been clarified in the text in lines 277-278 of the revised manuscript.

Line 275: Please state how many catchments were classified as "poor" and thus removed when the filter was applied.

This has been added in the text in lines 292-293. As seen in Figure 2, slightly more than half of the catchments have calibration NSE values below 0.7. This ranges from 84 to 89 basins depending on the hydrological model.

Line 307-308: "The twelve static descriptors presented in Table 1 allow the model to distinguish between each catchment.". Highlighting these variables in Table 1 may make it easier to understand which 12 are used as input to the LSTM. Also land cover (%) is split into 7 entries in Table 1 but I think is only considered as 1 of the 12 static descriptors which is confusing.

This was an error related to the fact that in our first simulations we were using 12 descriptors, until we found that using more (many from recommendations in the literature) allowed for better results. We

redid all the simulations but forgot to update this part of the text. It has been corrected in this revised version. All 25 catchment descriptors listed in Table 1 were used in this study.

Line 312: Please clearly define the training, validation, and testing catchments.

We forgot to clearly state the percentage of catchments used in training and validation in the methods and only referred to the values in line 408 of the original manuscript, where we explained that it is 80% of (N-1) basins used in training and 20% used in validation, with the pseudo-ungauged basin being the only one used as the "testing" basin. This is to ensure the LSTM model does not over-train/overfit, and the "regionalization" skill is evaluated on the completely independent testing basin. This has been clarified in lines 344-352 in the revised manuscript.

Line 330: Why was model #7 chosen as the LSTM structure of choice? Please state the reasoning.

In figure 8, we can see that the trend is monotonously increasing from model 1 to 7, in increasing complexity order. The median testing NSE increases from 0.740 for the simple model to 0.785 for the most complex model. Furthermore, each of the quantiles of the distributions are improving with each successive model. These types of improvements in regionalization are very significant. Therefore, the most complex model was selected since it outperformed the others, without requiring the modeller to integrate new physics/physical process representation. Simply by adding LSTM layers, the LSTM model was able to perform better in testing/regionalization mode, at the expense of computing time. It is true that building and training complex hydrological models, in the classical sense, requires a lot of effort. In this paper, we show that the performance can be better than what is obtained with hydrological models, but with little modelling effort from the hydrologist. Therefore, if the computing time is available, the more complex model is to be preferred, especially since the training is only performed once for a given ungauged catchment application. A section to this effect has been added to the text in lines 530-535.

Line 374-375: Were non-linear relationships between catchment descriptors and NSE values considered?

At this stage, no, only linear relationships were considered, to see if there was a correlation (i.e., if perhaps larger basins reacted better than smaller basins, etc.). However, this was not the case, leading to believe that the LSTM was able to use these descriptors in a non-linear fashion to provide the good, basin-dependent regionalized streamflow. This was clarified in line 409.

Line 395-396: "relatively simple LSTM model". Is this still referring to model #7 which is the most complex of the LSTM models tested? Please clarify. Also, on line 489 - "simple LSTM model".

Thank you for this comment. The text does indeed mention that it is a "simple LSTM model". However, this is relative, as in our opinion, an LSTM (even if the structure is internally quite complex) did not require much setting-up, calibrating, adjusting, etc. compared to other, more classical hydrological models. It is true that the LSTM model structure is quite complex compared to others, so this text has been modified to reflect this, i.e., that the LSTM model is complex but can be applied without a lot of work to represent the specific processes etc. The changes have been included in lines 430-432, and indirectly in lines 530-535.

Lines 455-459: As discussed in the introduction (lines 119-126) traditional hydrological models and LSTM models show different behaviours in terms of performance for increasing lengths of data (e.g., the plateauing after 3 years of the GR4J model (line 122)). Please comment on the "fair-ness" of the comparison considering only catchments with at least 30 years of data are included?

This is a fair point, and we added a discussion point to reflect on it in lines 501-511. In theory, both he GR4JCN and the LSTM model have access to the same data and as such, the comparison is as fair as it can be. However, GR4JCN must compromise on the parameter sets to use to be "generally" good, whereas the LSTM has many more degrees of freedom to fit to various hydroclimatological situations. However, GR4JCN has a predefined structure where processes are directly defined, whereas the LSTM must build its internal structure using its more numerous "parameters". We added a discussion point detailing the fact that not only the LSTM is to be favored due to the long time series of available data, but also that it can ingest data from many more catchments as well, whereas GR4J is limited to containing information from one catchment at a time.

Technical corrections

These technical corrections have also all be addressed in the revised version of the manuscript:

Line 12: Suggest changing "A series of …" to "a set of …" as series implies that there is a sequential element to the methods.

Line 12: "regionalization methods are applied"

Line 180-181: "Environment and Climate Change Canada (ECCC), and the United States Geological Survey (USGS)."

Line 232: Suggest changing "for each scenario" to "for each of the 18 scenarios" for clarity.

Line 288: "have difficulty remembering"

Line 315: "then converted from m.s-1 to mm.d-1" (as the division by drainage area would already have removed two spatial dimensions).

---

## Referee Report (RR1)

Review of

Continuous streamflow prediction in ungauged basins:
Long Short-Term Memory Neural Networks clearly outperform
traditional hydrological models

by Richard Arsenault, Jean-Luc Martel, Frédéric Brunet, François Brissette, and Juliane
Mai
[Paper #2022-295]

General evaluation

The authors provided satisfactory responses to most of my questions. They clarified the
utilization of LOOCV, and the training of the LSTM with the given dataset. The
upgraded literature review is well organized, yet they could have surveyed so many other
relevant references (from North America) addressing the same topic and not solely
focusing on those studies conducted by one specific research group. Thus, to further
improve the quality of the manuscript, I believe, there are still a few issues that need be
dealt with before the paper is deemed acceptable. Thus, for the time being, I would say
moderate revisions are necessary and required.

Comments/suggestions/recommendations

P4              On this page, the authors attempted to complete the literature review,
                the work done by Feng D, Lawson K, Shen C. (Prediction in
                ungauged regions with sparse flow duration curves and input-
                selection ensemble modeling. arXiv preprint arXiv:2011.13380.
                2020 Nov 26) should be cited since it is related to the concept
                brought in this study and has been applied in North America as well.

P11             Line 335, according to this sentence written by authors:
                « *Streamflow was not scaled itself using this approach, since the
                model outputs and target values are not part of the model training
                computations and thus have no impact on the numerical convergence
                efficiency*. ». This sentence raises two questions: (i) If the authors
                have not used the target values during the training, how do they train
                the model? Indeed, one of the essential elements to train and
                calibrate ML models is the target since the model cannot decipher
                the physics. (ii) I believe, the rationale behind the authors' work; that
                is the normalization of the target is not necessary, is not quite
                correct. Indeed, a target variable with a large spread of values, in
                return, may result in a large error gradient values; causing the weight
                values to change significantly, leading to an unstable learning

process and the occurrence of a rather slow learning process. Please correct the text accordingly to avoid any confusion.

P12      The response to the question about using Leaky ReLU was satisfactory. Yet, there is a need for additional clarification. If the LSTM structure used in this study is like the one proposed by Kratzert *et al*. (2018) (It appears that the authors just used two LSTM units to make a more complex one), the authors must mention the work done by Kratzert *et al*. when they are discussing the model structure. This way, more details will be provided to the readers to extract the information regarding model structure (e.g., hyperparameters).

P16      According to the following sentences, « *For example, in this study one catchment has a much larger area than almost all the others. For a hydrological model-based regionalization approach, this might skew the regressions between catchment descriptors and model parameters. LSTM, on the other hand, are strongly non-linear and are thus not bound to these limitations. They could also use these data to better predict streamflow processes at scales between the small and large catchments.* ». Is this where the authors explain the use of large catchments in LSTM modeling? Since I am not convinced with the explanations provided by the authors on how they rationalize including large catchments in their modeling. If that is because of the complexity of the model, it should be clarified thoroughly. Please provide more reasoning.

P17      In the part highlighted in blue, can the authors verify what do they want to explain to the readers? Since it is not quite clear which point (within the comments) is addressed here.

P28.      To avoid any confusion, please correct the information provided in the caption of Figure 3. It is mentioned that *N*-1 catchments are used during training which is not correct. These catchments are used during training and validating using an 80-to-20 split, respectively.

**Figures and Tables**

None, all the tables and figures are well organized.

**Editorial comments**

None, this is a well-written paper.

---

## Author Response (AR2)

We would first like to thank the Reviewers and Editor once again for this second review of our manuscript, which will again help improve this paper. As in the previous round, this document presents the modifications made to the original manuscript in response to the reviewer comments and suggestions, with the original comments in black font and author responses in blue.

**Reviewer #1**

Thank you for the recommendation and highlighting this typo, which has been fixed in the revised manuscript:

Line 338: "and converting to mm.d-1"

**Reviewer #2**

General comments

The authors provided satisfactory responses to most of my questions. They clarified the utilization of LOOCV, and the training of the LSTM with the given dataset. The upgraded literature review is well organized, yet they could have surveyed so many other relevant references (from North America) addressing the same topic and not solely focusing on those studies conducted by one specific research group. Thus, to further improve the quality of the manuscript, I believe, there are still a few issues that need be dealt with before the paper is deemed acceptable. Thus, for the time being, I would say moderate revisions are necessary and required.

We would like to thank Reviewer #2 for their comments and suggestions for improving the paper even further. We have provided a point-by-point reply to all issues and comments below.

Specific comments

P4:  On this page, the authors attempted to complete the literature review, the work done by Feng D, Lawson K, Shen C. (Prediction in ungauged regions with sparse flow duration curves and input selection ensemble modeling. arXiv preprint arXiv:2011.13380. 2020 Nov 26) should be cited since it is related to the concept brought in this study and has been applied in North America as well.

Indeed, for the literature review we stayed mostly with studies looking into continuous streamflow prediction rather than regionalization of hydrological indices, prediction (in the forecasting sense) and other such models. For this specific domain, we have not found any more pertinent studies. The reference recommended was however added in lines 98-99:

"Feng et al. (2020) showed that using regional flow duration curves as predictors in an LSTM model improved the prediction in ungauged basins skill over 671 CAMELS basins compared to an LSTM model without the flow duration curve inputs"

However, it is a preprint since 2020 and the paper refers to supporting information that does not seem to be available in the public domain. Therefore, we are unsure this reference will remain after the editing/typesetting stage. This is why it was not added at a prior stage.

P11 Line 335: according to this sentence written by authors: « Streamflow was not scaled itself using this approach, since the model outputs and target values are not part of the model training computations and thus have no impact on the numerical convergence efficiency. ». This sentence raises two questions: (i) If the authors have not used the target values during the training, how do they train the model? Indeed, one of the essential elements to train and calibrate ML models is the target since the model cannot decipher the physics. (ii) I believe, the rationale behind the authors' work; that is the normalization of the target is not necessary, is not quite correct. Indeed, a target variable with a large spread of values, in return, may result in a large error gradient values; causing the weight values to change significantly, leading to an unstable learning process and the occurrence of a rather slow learning process. Please correct the text accordingly to avoid any confusion.

Sorry for not clarifying sufficiently. First, yes, streamflow is used for training. This is indeed the entire basis of the training and very much required to obtain the model hyperparameters used for regionalization to the ungauged basins. This has been clarified in the text.

Second, the normalization aspect can be a problem in some cases when the target variable spans multiple orders of magnitude and can cause convergence issues. This was not a problem (and thus not required) in this study for a few reasons. First, the LSTM model was trained using scaled streamflow (streamflow divided by catchment area). This means that the range is already quite reduced and well outside the range of magnitudes that would cause problems in the gradient estimation / exploding gradients. Second, the learning rate was dynamically adjusted to progressively diminish every 5 epochs such that the convergence was controlled the entire time. Therefore, the normalization of streamflow was not required in this study.

This was clarified in the text as follows in lines 342-348:
"The target variable of streamflow was not itself scaled using this approach, since the model output and target values are not part of the model training computations and thus have no impact on the obtained results. Instead, the specific streamflow was used as the target variable by dividing streamflow records by the drainage area and converting to mm.d-1. This was done to allow combining information from the multiple training catchments during the LSTM training since all streamflow values were now represented in an area-independent depth unit, while at the same time ensuring all values had similar magnitudes to avoid convergence problems."

And the learning rate was also detailed in lines 365-366:
"In all cases, a decaying learning rate was implemented to ensure proper convergence of the training algorithm, refining the learning rate as a function of the number of epochs."

P12 The response to the question about using Leaky ReLU was satisfactory. Yet, there is a need for additional clarification. If the LSTM structure used in this study is like the one proposed by Kratzert et al. (2018) (It appears that the authors just used two LSTM units to make a more complex one), the authors must mention the work done by Kratzert et al. when they are discussing the model structure. This way, more details will be provided to the readers to extract the information regarding model structure (e.g., hyperparameters).

Thanks for this suggestion. We have added information regarding the Kratzert et al. (2019) setup in lines 323-328:

"In their paper, Kratzert et al. (2019b) tested multiple model structures and hyperparameters, including up to 256 units per LSTM layer, both for one and two layers, and with dropout rates ranging from 0.0 to 0.5 and input sequence lengths of 90 to 365 days. They finally settled for the model that provided the highest median, which was a single-layer, 256-unit LSTM with a dropout rate of 0.4 and an input sequence length of 270 days. However, the static descriptors were directly embedded in the LSTM layers, as opposed to their addition in a separate, parallel branch that is also tuned during training in this study."

P16 According to the following sentences, « For example, in this study one catchment has a much larger area than almost all the others. For a hydrological model-based regionalization approach, this might skew the regressions between catchment descriptors and model parameters. LSTM, on the other hand, are strongly non-linear and are thus not bound to these limitations. They could also use these data to better predict streamflow processes at scales between the small and large catchments. ». Is this where the authors explain the use of large catchments in LSTM modeling? Since I am not convinced with the explanations provided by the authors on how they rationalize including large catchments in their modeling. If that is because of the complexity of the model, it should be clarified thoroughly. Please provide more reasoning.

Yes indeed, this is where we justify the added value of large catchments in the dataset, and we have clarified further the text as follows, in lines 484-488:

"This is because neural networks in general, including LSTM-based neural networks, are particularly good for interpolating within the domain they are trained to represent but can be unpredictable while extrapolating outside of the parameters of their training dataset. Therefore, adding catchments with a wide array of properties confers the ability to establish relationships that other methods simply cannot attain by widening the domain on which the model can interpolate."

P17 In the part highlighted in blue, can the authors verify what do they want to explain to the readers? Since it is not quite clear which point (within the comments) is addressed here.

This point was in response to a previous comment (comment for lines 455-459 of Reviewer #2 in the previous review round) regarding the length of available data being a factor in the obtained results. It would be expected that a classical hydrological model would fare better than an LSTM if only 2 or 3 years of data were available, for example, because the hydrological model has a priori knowledge of the expected physics while the LSTM would need to build that model itself from the data. However, a hydrological model can use only the data from the catchment itself, whereas the LSTM can learn from other catchments as well, meaning it can also perhaps work well on a series of catchments that have few years of data each, if there are enough such catchments.

We have clarified this in the text in lines 510-520 of the revised manuscript.

P28. To avoid any confusion, please correct the information provided in the caption of Figure 3. It is mentioned that N-1 catchments are used during training which is not correct. These catchments are used during training and validating using an 80-to-20 split, respectively.

Thanks for highlighting this discrepancy. It has been corrected in the text as follows:

"Performance of hydrological models calibrated at each of the 148 study basins individually against the performance of the LSTM model in leave-one-out cross-validation (LOOCV) where the ungauged basin in question is not included in the set of basins used to train the LSTM and where the LSTM is trained and validated on 80% and 20% of the gauged basins, respectively."